# PIMRL: Physics-Informed Multi-Scale Recurrent Learning for Spatiotemporal Prediction

## Abstract

Simulation of spatiotemporal systems governed by partial differential equations is widely applied in fields such as biology, chemistry, aerospace dynamics, and meteorology. Traditional numerical methods incur high computational costs due to the requirement of small time steps for accurate predictions. While machine learning has reduced these costs, long-term predictions remain challenged by error accumulation, particularly in scenarios with insufficient data or varying time scales, where stability and accuracy are compromised. Existing methods often neglect the effective utilization of multi-scale data, leading to suboptimal robustness in predictions. To address these issues, we propose a novel multi-scale learning framework, namely, the Physics-Informed Multi-Scale Recurrent Learning (PIMRL), to effectively leverage multi-scale data for spatiotemporal dynamics prediction. The PIMRL framework comprises two modules: the micro-scale module embeds physical knowledge into neural networks via pretraining, and the macro-scale module adopts a data-driven approach to learn the temporal evolution of physics in the latent space. Experimental results demonstrate that the PIMRL framework consistently achieves state-of-the-art performance across five benchmark datasets ranging from one to three dimensions, showing average improvements of over 9% in both RMSE and MAE evaluation metrics, with maximum enhancements reaching up to 80%.

## 1 Introduction

In the field of natural sciences, physical systems governed by partial differential equations (PDEs) have found widespread applications across disciplines including biology, chemistry, meteorology, etc. (Anderson & Wendt, 1995; Blazek, 2015; Moukalled et al., 2016; Karniadakis & Sherwin, 2005; Zienkiewicz et al., 2005). Although numerical methods have been regarded as reliable tools for modeling these systems, the use of Direct Numerical Simulation (DNS) faces significant hurdles due to inherent limitations. DNS necessitates high spatial resolution and fine time stepping, resulting in considerable computational demands and prolonged processing times. A case in point is the simulation of aerodynamic flows around aircraft, which typically requires the generation of millions of grid points, thereby imposing prohibitive computational requirements (Ahmad, 2013; Goc et al., 2021). Moreover, these numerical simulation methods require complete physical prior knowledge, such as PDE formula, parameters, and initial/boundary conditions (Ferziger et al., 2019).

The ongoing development of artificial intelligence (AI) has propelled research into data-driven simulation methods, showcasing significant potential (Lu et al., 2021; Li et al., 2020; Stachenfeld et al., 2021). These methods, which do not require prior physics knowledge, offer user-friendly solutions. They also overcome traditional constraints on resolution and small time stepping, ensuring accurate solutions. Nevertheless, when faced with sparse data and multi-scale temporal challenges, these methods often struggle to optimally utilize data. They may either discard micro-scale data in favor of macro-scale data or vice versa, leading to compromised accuracy.

Multi-scale Burst sampling is a technique capturing multiple samples at a high rate over a short period of time to record rapidly changing events or transient phenomena, e.g., fast dynamics, followed by a low sampling rate to capture slow dynamics (see Figure 1). Compared to traditional continuous sampling methods, it provides high-resolution data at critical moments while maintaining resource efficiency. But conventional uniform-scale models struggle to fully utilize such data. The latent

ODE approach (Rubanova et al., 2019) may work for multi-scale sampled data, but remains unclear whether it can address PDE problems. There is an urgent need for a new method to handle such a type of data for spatiotemporal systems.

Data-driven methods face significant limitations when addressing challenges related to insufficient data and multiple time scales mentioned above. To overcome these obstacles, current approaches integrate physical knowledge into the model learning process. Specifically, physics-informed neural networks (PINNs) (Raissi et al., 2019a; 2020; 2019b) design initial and boundary conditions as penalty terms in the loss function, thereby leveraging physical laws in a "soft" way. However, this soft embedding approach can sometimes lead to unsatisfactory results. To some extent, the embedding methods used in PINNs are unable to ensure that the model fully adheres to the embedded physical conditions.

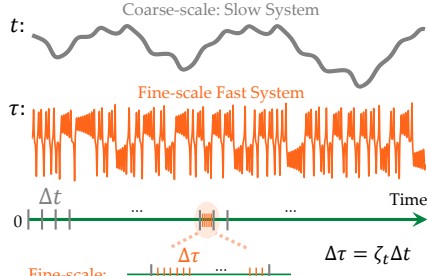

Figure 1: Multi-scale sampling, where $\Delta\tau$ denotes the micro-scale time interval for fast dynamics, $\Delta t$ the macro-scale time interval for slow dynamics, and $\zeta_t$ the scale separation variable (typically $\zeta_t < 1$ or $\zeta_t \ll 1$).

Directly embedding physical equations into the model architecture, as seen in the physics-encoded recurrent convolutional neural network (PeRCNN) (Rao et al., 2023), ensures strict adherence to the underlying physical laws. This approach addresses the issue of soft constraints in PINNs and enhances the model's interpretability and generalization capabilities. However, this approach requires continuous micro-scale data, which imposes stringent demands on the dataset and can introduce instability in long-term predictions. Alternatively, methods such as the learned interpolation (LI) model (Kochkov et al., 2021) and multiscale simulation frameworks (Vlachas et al., 2022) integrate numerical techniques with neural networks. These hybrid approaches enhance simulation efficiency while maintaining acceptable accuracy, although they still have some of the limitations inherent in traditional numerical methods.

Moreover, we observe that the error increases along with the prediction horizon, a phenomenon that is widespread. Although existing efforts are predominantly placed on enhancing the model's predictive capability, long-term prediction of spatiotemporal dynamics still suffers from error accumulation. Hence, properly controlling the accumulation of errors is becomes crucial. Based on this core idea, we are motivated to leverage comprehensive multi-scale data through a novel Physics-Informed Multi-Scale Recurrent Learning (PIMRL) framework to tackle the problems mentioned above. This framework consists of macro-scale and micro-scale modules. The micro-scale module, pretrained to learn the underlying physical laws, enhances the accuracy of simulations. The macro-scale module reduces the accumulation of errors by minimizing the number of rollout iterations for the micro-scale module, thereby enhancing long-term predictive performance. The main contributions of this paper are summarized as follows:

- We proposed a new PIMRL framework that effectively leverages information from multi-scale data for long-term spatiotemporal dynamics prediction. The concept of reducing the accumulation of errors is achieved through the integration of the macro-scale and micro-scale modules.

- We designed a novel message passing mechanism between micro- and macro-scale modules, which effectively transmits physical information, enhances the micro-module's correction capability, and reduces the number of micro-scale rollout iterations through the macro-scale module.

- The PIMRL model achieved optimal performance in effectively predicting the duration of multiple different cases from fluid dynamics to physical systems, demonstrating its scalability and laying a solid foundation for more generalizable models in the future.

## 2 RELATED WORK

Simulation tasks often aim to solve partial differential equations (PDEs) accurately and efficiently. Previous researchers have developed numerical methods achieving high precision with many nodes and short time steps. To speed up simulations, deep learning techniques have evolved from apply-

ing conventional algorithms to designing new models that integrate physical knowledge, including hybrid methods and models embedding physical principles.

**Computational fluid dynamics.** Computational Fluid Dynamics (CFD) is a branch of fluid mechanics that uses numerical methods and algorithms to analyze and solve problems involving fluid flow (Anderson & Wendt, 1995; Blazek, 2015; Moukalled et al., 2016; Karniadakis & Sherwin, 2005; Zienkiewicz et al., 2005). When faced with complex problems, unacceptable time and computational costs are the primary obstacles limiting the application of CFD. Moreover, when the corresponding physical equations contain unknown parameters or even unknown terms, numerical methods are unable to perform accurate simulations. The aforementioned methods have inspired us to leverage physical knowledge. However, to overcome the aforementioned issues, we have decided to introduce deep learning methods into our PIMRL framework.

**Deep learning methods.** With the advancement of AI, the application of AI in physical system simulation tasks has become more diverse and profound. For example, classical convolutional neural network models (Stachenfeld et al., 2021; Bar-Sinai et al., 2019) and ResNet (Lu et al., 2018), U-Net (Gupta & Brandstetter, 2023) what originally used for image segmentation, graph neural networks (Sanchez-Gonzalez et al., 2020; Pfaff et al., 2020), and transformer-based models (Wu et al., 2024; Hang et al., 2024; Janny et al., 2023; Li et al., 2024) are now being employed. Neural operators used for learning mappings between function spaces have also seen significant development in physical simulation tasks, such as DeepONet (Lu et al., 2021), MWT (Gupta et al., 2021), FNO (Li et al., 2020; Tran et al., 2021; Rahman et al., 2022; Wen et al., 2022), etc. In addition, there are several methods specifically designed for spatiotemporal prediction tasks, such as ConvLSTM (Shi et al., 2015), PredRNN (Wang et al., 2022), and TrajGRU (Shi et al., 2017). The traditional deep learning methods mentioned above all require sufficient data for training; insufficient data can lead to poor model performance and severe overfitting issues. Moreover, error accumulation is another inevitable problem that limits the application of traditional deep learning methods. These limitations restrict the effectiveness and reliability of such models, especially in scenarios requiring long-term predictions or when data is scarce. The challenges outlined above have led us to propose a framework that effectively controls error accumulation and adeptly processes multi-scale data.

**Physics-informed deep learning methods.** To incorporate physical information into models, researchers have devised various methods. One category includes physics-inspired methods like Phy-CRNet (Ren et al., 2022), PINN (Raissi et al., 2019a), and PhySR (Ren et al., 2023). Another category involves physical embedding methods, where explicit physics knowledge is embedded into the model to fully leverage physical principles, like EquNN (Wang et al., 2020) and PDE-Net (Long et al., 2018; 2019). PeRCNN (Rao et al., 2023; 2022) offers a "hard" encoding mechanism to learn the dynamics of physical systems from limited data. This approach leverages prior physics knowledge for predictions, thereby equipping strong predictive capabilities and robust generalization across different initial conditions. However, all the aforementioned methods suffer from error accumulation, making it difficult to obtain stable and accurate results in long-term prediction tasks. PeRCNN effectively utilizes physical information in a way that can be leveraged within PIMRL.

**Hybrid learning methods.** In recent years, an emerging research direction has been to integrate deep learning methods with traditional numerical methods. By combining classical solvers with deep neural networks, hybrid approaches designed on this principle can achieve acceptable accuracy while operating faster than pure numerical methods. Examples include the Learned Interpolation (LI) model (Kochkov et al., 2021) and numerical discretization learning (Zhuang et al., 2021). However, in these hybrid methods, the numerical method component is not involved in the training process, and such approaches also require substantial amounts of data. The pioneering work multi-scale simulations of complex systems (Vlachas et al., 2022) employs a multi-scale framework for predictions, effectively reducing the required time and computational cost while maintaining good accuracy. However, this method still requires ample data for training. The idea behind hybrid learning methods is very promising, so we have introduced our own message-passing mechanism within our PIMRL framework.

## 3 METHODOLOGY

We propose the PIMRL framework, as illustrated in Figure 2, for spatiotemporal dynamics prediction with a small amount of multi-scale training data

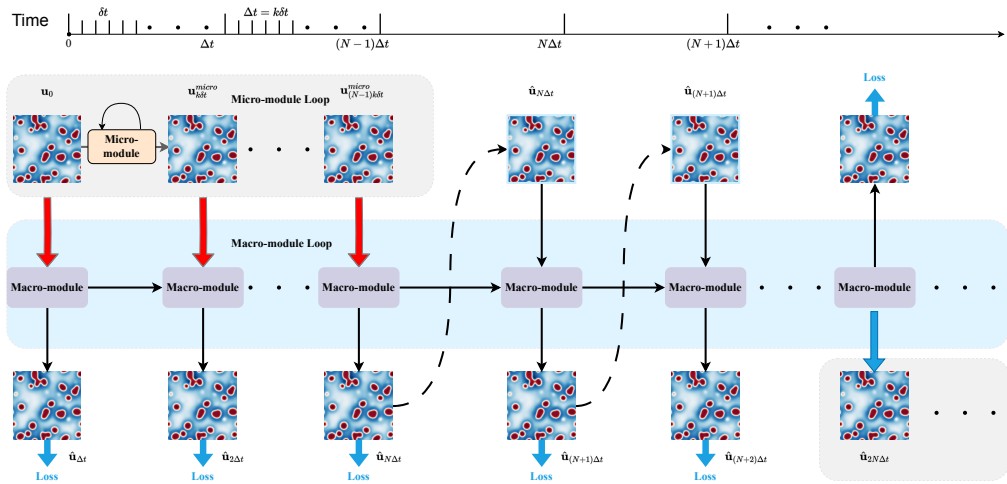

Figure 2: The overall framework architecture, which integrates physics-informed constraints with deep learning. The initial state of the system is denoted by $\mathbf{u}_0$. The state predicted by the micro-module after $k$ iterations, where each iteration occurs at intervals of $\delta t$, is represented by $\mathbf{u}_{k\delta t}^{micro}$. The predicted value of the physical state from PIMRL is denoted by $\hat{\mathbf{u}}$.

### 3.1 PARTIAL DIFFERENTIAL EQUATIONS SIMULATION TASK

Simulation tasks are intimately connected with PDEs, which are essential for describing and simulating physical models, and the time-dependent PDEs are defined as:

$$\mathbf{u}_t = \mathcal{F}(t, x, \mathbf{u}, \nabla\mathbf{u}, \mathbf{u} \cdot \nabla\mathbf{u}, \nabla^2\mathbf{u}, \cdots; \mu), \tag{1}$$

where $\mathbf{u}(x, t)$ denotes the spatiotemporal solution field, $\mathbf{u}_t$ the first-order time derivative, $\mathcal{F}(\cdot)$ a linear/nonlinear function, $\nabla$ the Nabla operator, $\nabla^2$ the Laplace operator, and $\mu$ the PDE parameter.

In addition, we define the initial and boundary conditions (ICs, BCs) for this equation, namely: $I[\mathbf{u}](x, t = 0) = 0$ and $B[\mathbf{u}](x, t) = 0$ where $I$ and $B$ indicate IC and BC operators.

### 3.2 OVERVIEW

As outlined in the introduction, we have designed the PIMRL framework to mitigate error accumulation while ensuring that it can effectively capture the intrinsic changes in the physical system, rather than disregarding the physical information over large time intervals. Additionally, PIMRL is capable of efficiently utilizing multi-scale data. Below, we will provide a detailed exposition of the PIMRL framework's architecture, the corresponding training methodologies, the macro and micro modules, as well as the boundary padding method that incorporates boundary conditions.

### 3.3 FORECASTING ARCHITECTURE

As shown in Figure 2, the PIMRL framework comprises two key components: the micro-scale module and the macro-scale module. The message-passing mechanism is an interaction between the micro-module and the macro-module. The information transmitted includes physical knowledge learned by the micro-module and corrections applied to the macro-module. Additionally, the macro-module also passes information to future iterations of the micro-module.

PIMRL is designed to achieve long-term prediction of spatiotemporal dynamics. PIMRL operates in a recursive manner, involving cycles at the micro-module level, the macro-module level, and the overall framework level. Only the output from the macro-module contributes to the final output of the PIMRL framework and is used to compute the loss for training the entire framework, whereas the output from the micro-module serves as intermediate variables within the framework. And the message-passing mechanism is described as follows:

Figure 3: PIMRL includes two main modules: (a) the micro-module, designed to capture local features and small-scale dynamics; and (b) the macro-module, which captures long-term dependencies and global patterns using residual connections.

- Firstly, the micro-module loop is a simple autoregressive process with time step $\delta t$, where the output at the previous time step serves as the input for the next time step. Secondly, the macro-module loop performs self-cycles.

- When the micro-module is involved in the prediction, for every $k$ steps of micro-module with $\delta t$ like Equation 3, the final output of micro-module is passed to the macro-module, and at this point, the output from the macro-module serves as the output of the entire PIMRL model shown as Equation 4. When the micro-module is not involved in the prediction, the macro-module loop is a simple autoregressive process with time step $\Delta t$.

- Finally, there is a PIMRL loop that operates in conjunction with the macro-module loop. After every $N-1$ cycles of the macro-module loops, the micro-module stops participating in the prediction, and the macro-module performs $N$ steps of autoregressive prediction on its own. This completes a total of $2N$ cycles. Each output from the macro-module during these $2N$ cycles serves as the output of PIMRL as depicted in Equation 2.

We utilize two equations to represent the relationship between the micro-scale module and the macro-scale module, denoted as $F_{micro}$ and $F_{macro}$ respectively, as illustrated in Figure 2. The details of this process can be represented using the aforementioned symbols as follows:

$$\text{PIMRL Loop: } \hat{\mathbf{u}}_{t+2N\Delta t} = \underbrace{F_{macro}(...F_{macro}}_{\times N}(\hat{\mathbf{u}}_{t+Nk\delta t})), \tag{2}$$

$$\text{Macro-module Loop: } \mathbf{u}_{k\delta t}^{micro} = \underbrace{F_{micro}(...F_{micro}}_{\times k}(\mathbf{u}_t)), \tag{3}$$

$$\text{Macro-module Loop: } \hat{\mathbf{u}}_{t+2k\delta t} = F_{macro}(\mathbf{u}_{k\delta t}^{micro}), \tag{4}$$

where the variable $\mathbf{u}$ denotes the physical state. The relationship between the micro-time step $\delta t$ and the macro-time step $\Delta t$ is given by $\Delta t = k\delta t$, where $k$ is an adjustable parameter determined by the time stepping of different scales in the real data.

### 3.4 TRAINING STRATEGIES

Since the PIMRL model consists of micro- and macro-scale modules, adopting brute-force end-to-end training yields unsatisfactory results (see the ablation study). Firstly, we establish a pre-training phase where only the micro-module is trained. The purpose of this pre-training phase is to enable the micro-module to effectively learn the dynamics of the physical system and the underlying physical laws, free from the influence of the macro-module. In the next phase, referred to as the overall training phase, all modules within the PIMRL framework are engaged in training together. During this phase, the micro-module benefits from the parameters pre-trained in the pre-training phase, serving to supervise and correct the macro-module. The output from the macro-module serves as the final output of the entire PIMRL framework and is used to compute the loss. The details are shown in Appendix C.1.

### 3.5 MICRO-SCALE MODULE

The micro-scale module is designed to learn underlying physical laws that govern the spatiotemporal dynamics from micro-scale data with small time stepping, where we adopt the PeRCNN model (Rao

et al., 2023) with the architecture of $\Pi$-block shown in Figure 3(a). In a forward Euler scheme: $\mathbf{u}_{(k+1)\delta t} = \hat{\mathcal{F}}(\mathbf{u}_{k\delta t}) \cdot \delta t + \mathbf{u}_{k\delta t}$, where $\delta t$ denotes that the module predicting in micro-scale time stepping. We can then approximate the $\mathcal{F}$ by $\hat{\mathcal{F}}$ described as follows:

$$\hat{\mathcal{F}}(\mathbf{u}_{k\delta t}) = \sum_{c=1}^{N_c} W_c \cdot \left[ \prod_{N_l}^{l=1} (K_{c,l} \star \hat{\mathbf{u}}_{k\delta t} + b_l) \right]. \tag{5}$$

where $N_c$ denotes the channel count, and $N_l$ the total number of parallel convolutional layers. The symbol $\star$ denotes the convolutional operation. For each layer $l$ and channel $c$, $K_{c,l}$ designates the specific filter weight, while $b_l$ stands for the bias term of that layer $l$. In the context of a $1 \times 1$ convolutional layer, $W_c$ denotes the weight assigned to the $c^{\text{th}}$ channel, with the bias term being omitted here for the sake of simplicity and brevity.

When a certain term in the governing PDE remains known (e.g., the diffusion term $\Delta \mathbf{u}$), its discretization can be directly embedded in PeRCNN (called the physics-based Conv layer as shown in Figure 3(a)). The convolutional kernel in such a layer can be set according to the corresponding finite difference (FD) stencil. In essence, the physics-based Conv connection is constructed to incorporate known physical principles, whereas the $\Pi$-block is aimed at capturing the complementary unknown dynamics. The details of the physics-based FD Conv are provided in Appendix B.

### 3.6 MACRO-SCALE MODULE

The design of the macro-scale module, as a pivotal component of the PIMRL framework, is meticulously crafted to effectively manage and analyze macro-scale data. This type of data often poses unique challenges due to the substantial time spans it encompasses, which in turn lead to significant variations in the underlying physical states captured within the data points. These variations might be highly nonlinear and dynamic, making it difficult for traditional, physically motivated modeling methods to accurately capture all the nuances and complexities involved. As depicted in Figure 3, our macro-scale module utilizes ConvLSTM block with a residual connection. The structure of each block, illustrated in Figure 3(b), consists of a pair of encoders and decoders, along with a ConvLSTM cell shown in Appendix Figure S3. The input to this block undergoes mapping by the encoder, where the feature map serves as the characteristic of the latent space. Following this, the ConvLSTM cell simulates the dynamics, and the output is mapped back to the physical space through the decoder. Finally, the output, as a residual, is added to the feature from the previous time step to generate the prediction for the next time step. More details are provided in Appendix C.1.

### 3.7 BOUNDARY CONDITION PADDING

Inspired by PeRCNN (Rao et al., 2023), we introduce BC hard encoding in both micro-scale and macro-scale modules. This encoding method ensures that the feature maps comply with the given BCs during the convolution process while also serving the purpose of padding, which involves filling the feature maps before convolution operations. Specifically, in this paper, our case adheres to periodic BCs, and the application of the corresponding padding is illustrated in Figure 4. This encoding scheme ingeniously incorporates BCs into the padding, thereby enhancing the accuracy of the prediction.

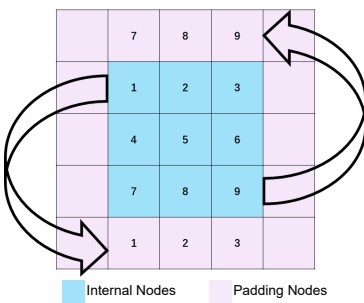

Figure 4: Periodic BC padding.

## 4 EXPERIMENTS

To validate the effectiveness and versatility of our proposed PIMRL framework, we conducted extensive experiments on a diverse set of fluid dynamics and reaction-diffusion systems equations. Specifically, we tested our model on the following cases: the 1D Korteweg-de Vries (KdV) equation, the 2D Burgers equation, and three reaction-diffusion (RD) equations. These equations represent a range of physical phenomena with varying degrees of complexity and nonlinearity. Our results show that PIMRL consistently outperforms existing methods in terms of accuracy and robustness across these challenging cases.

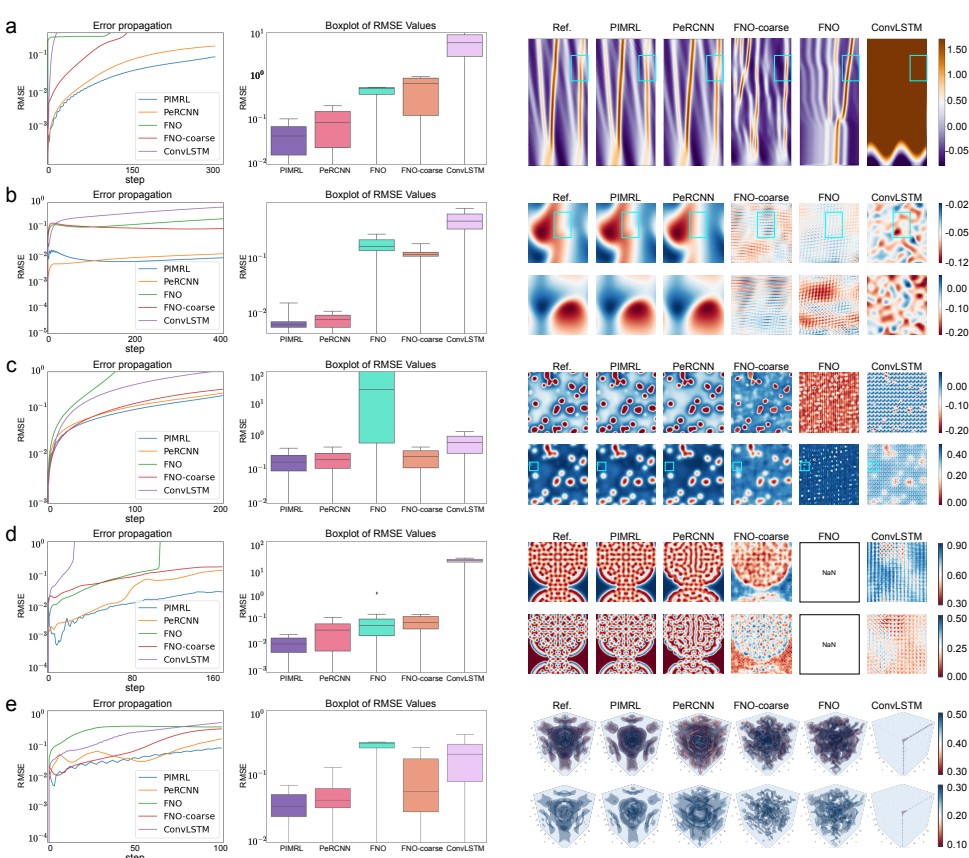

Figure 5: An overview of the comparison between our PIMRL framework and baseline models is provided, including error propagation curves (left), error boxplots (middle), and final prediction plots (right). Figures (a) through (e) respectively showcase the qualitative results for the KdV, Burgers, 2D GS, FN, and 3D GS cases.

**Datasets.** We conducted experiments on five datasets, including: Korteweg-de Vries (KdV), 2D Burgers, FitzHugh-Nagumo (FN), 2D Gray-Scott (2D GS), and 3D Gray-Scott (3D GS). The IC for the Kdv equation is created by summing multiple sine waves with random amplitudes, phases, and frequencies, resulting in a complex waveform. ICs for the Burgers' equation are generated randomly according to a Gaussian distribution. The FN equation is initialized with random Gaussian noise for a warm-up period, after which time sequences are extracted to form the dataset. The GS equation starts the reaction from random initial positions and then diffuses. In these cases, except for the Kdv case, which was solved using the Finite Volume Method (FVM), the rest were solved using the Finite Difference Method (FDM). Additionally, We have two sets of data with different time scales originating from the same ICs. The micro-scale data $\mathbf{U}^{\text{micro}} = \{\mathbf{u}_0, \mathbf{u}_{\delta t}, \mathbf{u}_{2\delta t} \cdots \} \in \mathbf{R}^{\text{micro}}$ is characterized by short and scattered continuous time intervals, while the macro-scale data $\mathbf{U}^{\text{macro}} = \{\mathbf{u}_0, \mathbf{u}_{\Delta t}, \mathbf{u}_{2\Delta t} \cdots \mathbf{u}_{\text{Tend}}\} \in \mathbf{R}^{\text{macro}}$ exhibits persistent continuity until the end. The validation and test sets are established based on different ICs but with the same parameters, making it more challenging than extrapolation under the same ICs. The dataset configuration is presented in Appendix G.

**Model training.** The primary objective is to first pretrain the micro-scale module using the micro-scale data to learn the underlying physics. Subsequently, the pretrained micro-scale module is integrated into the overall framework, and the entire model is trained using macro-scale data to capture the spatial evolution patterns over long time stepping. Both the pretraining and training processes can be formulated as auto-regressive rollout problems. The loss function is defined as $L(\boldsymbol{\theta}) = \frac{1}{BN} \sum_{i=1}^{B} \sum_{j=1}^{N} (y_{i,j} - \hat{y}_{i,j})^2$, where the $B$ and $N$ denote the number of batches and the batch size. The $\boldsymbol{\theta}$ indicates all the trainable parameters and $(y_{i,j} - \hat{y}_{i,j})$ means the difference be-

tween the rollout-prediction $\hat{y}_{i,j}$ of $j$-th sample in the $i$-th batch and the corresponding labeled data $y_{i,j}$. More training details are shown in Appendix E.

**Baseline models.** To validate the effectiveness of the proposed PIMRL framework, we introduced several baseline models. Firstly, we considered the widely recognized high-performing data-driven model FNO (Li et al., 2020), which has been trained on datasets with two different time intervals, denoted as FNO (trained on fine-scale data with small time steps) and FNO-coarse (trained on coarse-scale data with large time steps). In particular, we examined the impact of cumulative error on long-term predictions. Our results show that the influence of error accumulation is significant. Detailed analysis is provided in Section 4.1. Secondly, we included the PeRCNN model (Rao et al., 2023), which embeds physical knowledge in a hard way and demonstrates excellent performance on multiple datasets. Due to constraints on the time stepping of the model, PeRCNN is trained only on datasets with small time stepping. Lastly, we incorporated ConvLSTM (Shi et al., 2015), a classic sequential prediction model trained on datasets with large time stepping, serving as the macro-scale module within our framework in a residual way. The details are shown in the Appendix E.

**Evaluation Metrics.** To comprehensively evaluate the performance of our model, we adopted several metrics: Root Mean Square Error (RMSE), Mean Absolute Error (MAE), and High Correction Time (HCT). These metrics provide a multi-faceted assessment of the model's accuracy and reliability. RMSE is calculated on the macro-scale data to facilitate comparisons between models operating at different granularities. MAE provides a measure of the average absolute difference between the predicted and actual values,less sensitive to outliers compared to RMSE. HCT evaluates the time it takes for the model to correct its predictions to a high level of accuracy, which is particularly important for long-term prediction. Detailed formulas for these metrics are provided in Appendix F.

## 4.1 MAIN RESULTS

Figure 5 illustrates the performance results of our framework in comparison with various baseline models across different cases, including error propagation, showcasing box plots of different quantities and their prediction outcomes. Additionally, Table 1 presents relevant quantitative metrics as an expression of the results, where $*$ in Table 1 indicates that the inference process has reached the end of the test data.

**KdV Equation.** The KdV equation describes the evolution of nonlinear wave phenomena. In this study, due to the absence of one-dimensional instances in the PeRCNN model, we designed our own physical embedding module and corresponding model, following the principles of PeRCNN as detailed in Appendix B. We observed significant discrepancies between the predictions of the PeRCNN model and the ground truth, as shown in the blue box of Figure 5(a). Meanwhile, the other baseline models performed poorly at the start, likely due to the complexity of the KdV equation. In contrast, our proposed model demonstrated substantial advantages over the baseline models in terms of predictive accuracy. It maintained a consistent basic shape with the ground truth values, even in long-term forecasts. As shown in Table 1, the PIMRL framework achieved 30% to 50% improvements in the evaluation metrics, highlighting a significant advancement in the field.

Table 1: Quantitative results of our model and baselines.

| Case | Model | RMSE ↓ | MAE ↓ | HCT (s) ↑ |
|------|-------|--------|-------|-----------|
| KdV | ConvLSTM | 5.8507 | 7.6036 | 9.6 |
| | FNO | 0.4891 | 0.3300 | 0.45 |
| | FNO-coarse | 0.5461 | 0.4167 | 7.8 |
| | PeRCNN | 0.0942 | 0.0941 | 30 |
| | PIMRL(Ours) | **0.0457** | **0.0607** | **46.2** |
| | Promotion (↑) | 51.5% | 35.5% | 54.0% |
| Burgers | ConvLSTM | 0.4020 | 0.3232 | 0.176 |
| | FNO | 0.1561 | 0.1301 | 0.104 |
| | FNO-coarse | 0.1094 | 0.0879 | 0.064 |
| | PeRCNN | 0.0075 | 0.0058 | 3.216 |
| | PIMRL(Ours) | **0.0068** | **0.0049** | **3.216** |
| | Promotion (↑) | 9.3% | 15.6% | 0%* |
| FN | ConvLSTM | 0.5077 | 0.4925 | 1.86 |
| | FNO | 937 | 2393980 | 1.65 |
| | FNO-coarse | 0.1878 | 0.1643 | 4.98 |
| | PeRCNN | 0.1591 | 0.1139 | 6.99 |
| | PIMRL(Ours) | **0.1349** | **0.0990** | **7.74** |
| | Promotion (↑) | 15.2% | 13.1% | 10.7% |
| 2DGS | ConvLSTM | 15.7559 | 13.7966 | 195 |
| | FNO | NaN | NaN | 810 |
| | FNO-coarse | 0.0884 | 0.0629 | 1335 |
| | PeRCNN | 0.0455 | 0.0268 | 1379.5 |
| | PIMRL(Ours) | **0.0133** | **0.0072** | **1965*** |
| | Promotion (↑) | 70.8% | 73.1% | 42.5% |
| 3DGS | ConvLSTM | 0.2081 | 0.2009 | 562.5 |
| | FNO | 0.2798 | 0.1950 | 112.5 |
| | FNO-coarse | 0.1042 | 0.0611 | 360 |
| | PeRCNN | 0.0532 | 0.0977 | 510 |
| | PIMRL(Ours) | **0.0381** | **0.0190** | **731.25** |
| | Promotion (↑) | 28.4% | 80.6% | 43.4% |

**2D Burgers Equation.** On the right side of Figure 5(b), we observe that besides our framework, only PeRCNN achieved satisfactory results after training on the micro-scale, while the remaining baseline models failed to capture the physical changes effectively. Additionally, in the areas marked by the blue box in Figure 5(b), we can see errors in PeRCNN's predictions for long-term forecasts, whereas our PIMRL framework demonstrates superior performance, as also visually depicted in Figure 5(b). In the error propagation shown in Figure 5(b), we notice that although initially our framework exhibits higher RMSE compared to the PeRCNN model due to the macro-scale module, the accumulated errors lead to inaccurate predictions by PeRCNN over time. While PeRCNN and PIMRL exhibit the same performance on the HCT metric, this similarity arises from insignificant changes in the later stages. The cumulative error of PeRCNN does not reflect in this particular metric. As shown in Table 1, there are significant improvements in both RMSE and MAE. Additionally, the noticeable divergence of PeRCNN can be observed in the snapshot of the final step.

**2D FitzHugh-Nagumo Equation.** In the Figure 5(c), we can observe that the purely data-driven method mentioned earlier fails to fully exhibit its original performance on this relatively small dataset. FNO and FNO-coarse represent the effectiveness of the FNO method when trained on datasets of different granularities, respectively. It is worth noting that the performance of FNO-coarse surpasses that of the FNO model trained on micro-scale data. This result effectively validates our hypothesis that error accumulation can significantly impact the model's performance in long-term multi-step predictions. Quantitatively, our model has improved by at least 10% compared to the currently best-performing model. Furthermore, in the Table 1 we observe that the variation curves of RMSE and other metrics for our model and the PeRCNN model exhibit a striking similarity, yet our model outperforms PeRCNN. This not only indicates that PeRCNN effectively leverages its supervised refinement capabilities but also demonstrates its ability to mitigate the influence of cumulative errors through the evolution of larger time stepping.

**2D Gray-Scott Equation.** As shown in the left part of Figure 5(d), only PIMRL effectively carried out long-term predictions, showcasing a clear demonstration of the cumulative errors of PeRCNN in this case. The model exhibited significant deviations from ground truth values over extended periods, highlighting its limitations in capturing long-term variations. Among the other models, FNO-Coarse exhibited the best performance. It is evident that FNO performed poorly in the absence of physically informed embeddings, particularly with small time stepping. In this case, Table 1 further validates the superior performance of PIMRL with both RMSE and MAE showing an improvement of over 70%, and the HCT similarity of PIMRL still at 0.99 at the final prediction step. This underscores the effectiveness of the PIMRL framework in enhancing predictive accuracy and maintaining consistency in the predictions throughout the forecasting process. Such robust performance metrics highlight the potential of PIMRL in addressing complex data-driven challenges.

**3D Gray-Scott Equation.** As shown on the right side of Figure 5(e), the predictions under our framework closely align with the ground truth. Through box plots and error evolution graphs, it is evident that PeRCNN, as the best-performing model among the baselines in our study, outperformed the other models lacking physical knowledge embeddings, especially when compared to FNO trained on the same dataset. In quantitative analysis of Table 1, the improvements are also significant, not only in the overall evaluation metrics of RMSE and MAE in 28.4% and 80.6% but also in the substantial growth of the HCT in 43.4%.

## 4.2 Ablation Study

To evaluate the impact of components of PIMRL and demonstrate the effectiveness of our framework structure, we have designed five novel models and provided their RMSE results in the FN case. (1) The ablation study with the "PIMRL w/o Connect", where the connections between the micro-module and the macro-module are re-moved, is designed to demonstrate the effectiveness of the PIMRL framework's structural design. This experiment, which leaves only the serial structure of the two modules, shows that the connections within the PIMRL framework are essential for its performance. (2) The "FNO-MRL" replaces the micro-scale module containing physical information, PeRCNN, with the data-driven model FNO, aiming to validate the efficacy of the physical embedding in the micro-scale module

Table 2: Results for ablation study.

| Model | RMSE |
|---|---|
| PIMRL w/o Connect | 0.1975 |
| FNO-MRL | 0.7854 |
| PIMRL w/o Pretraining | 0.2599 |
| PIMRL w/o Physics-based FD Conv | 0.1738 |
| PeRCNN w/o Physics-based FD Conv | NaN |

within our framework. (3) "PIMRL w/o Pretraining" provides a perspective on the training method by eliminating the pretraining step for the micro module. In this ablation study, the absence of pretraining leads to inferior performance compared to the PIMRL. This demonstrates that directly introducing large time intervals for training can deprive the micro module of the opportunity to learn fine-grained changes, similar to why PeRCNN cannot be directly used with large time intervals. (4) "PIMRL w/o Physics-based FD Conv" indicates the removal of the Physics-based FD Convolution. This ablation study emphasizes the effectiveness of the Physics-based FD Conv by showing the performance degradation when it is omitted. (5) "PeRCNN w/o Physics-based FD Conv" is a version of PeRCNN without the Physics-based Finite Difference (FD) Convolution. While this version can initially make relatively accurate predictions, the errors accumulate over subsequent iterations, eventually becoming unacceptably large.

The five ablation studies conducted not only validate the effectiveness of the interaction design by removing the connection mechanisms but also highlight the contribution of our proposed connection approach. Additionally, by substituting the micro-modules with non-physics-embedded data-driven models, the experiments confirm the efficacy of the physics-embedding within our PIMRL framework. Subsequent pre-training ablation experiments and the removal of the Physics-based FD Conv further substantiate the effectiveness of the corresponding methods and modules.

### 4.3 INFERENCE TIME

In the aforementioned cases, PIMRL not only achieves state-of-the-art performance in long-term predictions but also significantly reduces the computational time cost. Compared to traditional methods such as Direct Numerical Simulation (DNS), our framework is substantially faster, demonstrating a significant improvement in computational efficiency (given the same computing facility, aka, CPU). Additionally, PIMRL not only delivers superior pre-

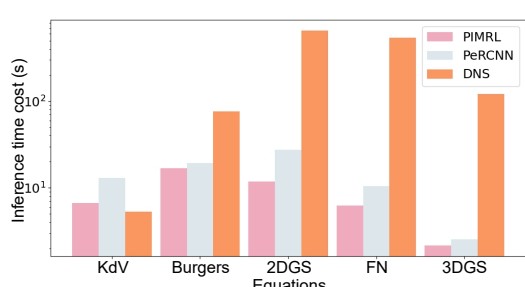

Figure 6: Computational time for comparison

dictive accuracy, but also outperforms PeRCNN in terms of computational efficiency. The numerical methods used for DNS are consistent with the parameter settings during data generation. The difference lies in the required prediction time length, which needs to be the same as that of PIMRL and PeRCNN for a fair comparison.

## 5 CONCLUSIONS

This paper introduces a new multi-scale temporal model named Physics-Informed Multi-Scale Recurrent Learning (PIMRL) for prediction of spatiotemporal dynamics. Adhering to the idea of controlling cumulative error by reducing iterations, PIMRL integrates modules across different scales within its framework to realize this concept, and it can efficiently leverage multi-scale data, which facilitates learning physical laws with limited resources. From fluid dynamics to reaction-diffusion systems, PIMRL demonstrates superior performance in handling multi-scale data, providing accurate long rollout predictions. Our future work will optimize the macro-scale model for time-series tasks to improve computational efficiency and predictive accuracy. We also plan to integrate super-resolution techniques to enhance the model's adaptability to various spatiotemporal scales, broadening its application to more complex physical systems.

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

APPENDIX

# A IMPACT STATEMENT

The paper endeavors to devise a adaptable framework that expedites simulations and predictive analyses of physical systems by utilizing multi-scale temporal data. This framework harmoniously integrates data-driven methodologies with physics-informed principles, striking a delicate balance between empirical insights and theoretical underpinnings in its application. The framework can be widely applied in various research fields including material science, turbulent flow prediction, chemical engineering, and so forth.Our research is exclusively conducted for the pursuit of scientific objectives and does not entail any potential ethical concerns or risks.

# B DESIGN OF THE PHYSICAL FILTER

In the one-dimensional problem KdV, the original paper of PeRCNN did not provide the corresponding model design. Following their concept, we present the corresponding physical filter.

$$f_{\text{xxx}} = \frac{-f_{i-2} + 2f_{i-1} + 0f_i - 2f_{i+1} + f_{i+2}}{2h^3} \tag{S1}$$

As shown in the Figure S1, we designed a Physical-Filter to represent $\frac{\partial^3 u}{\partial x^3}$ in the KdV equation. This

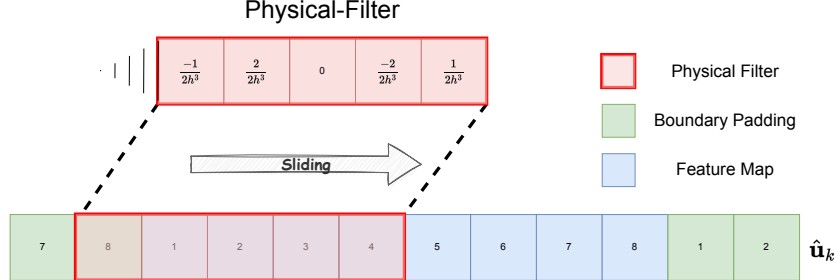

Figure S1: Filter for KdV

approach leverages the inherent physical properties of the system to accurately model the third-order spatial derivative, thereby enhancing the accuracy and efficiency of the numerical solution. Among the parameters, $h$ indicates the $\Delta x$ in the cases. The Boundary Padding is an approach to adapt to periodic boundary conditions by replacing the original zero padding with a periodic boundary padding.

# C IMPLEMENTATION DETAILS

## C.1 OVERVIEW

The overview of the PIMRL framework, which includes a pretraining stage using micro-scale data for physics-informed Learning and the utilization of a micro-module, informed by learned physics knowledge, to correct the macro-module during training on macro-scale data.

In the main body of this paper, we have elaborated on the architectures at the micro and macro scales. Within the macro-scale module, there are components including an encoder, a decoder, and a Residual Long Short-Term Memory (ResidualLSTM). The following section will provide a detailed exposition of their configuration specifics.

## C.2 ENCODER AND DECODER

In our framework, the autoencoder is employed not for the purpose of minimizing reconstruction loss. Instead, the encoder is utilized to extract features, while the decoder serves to project the

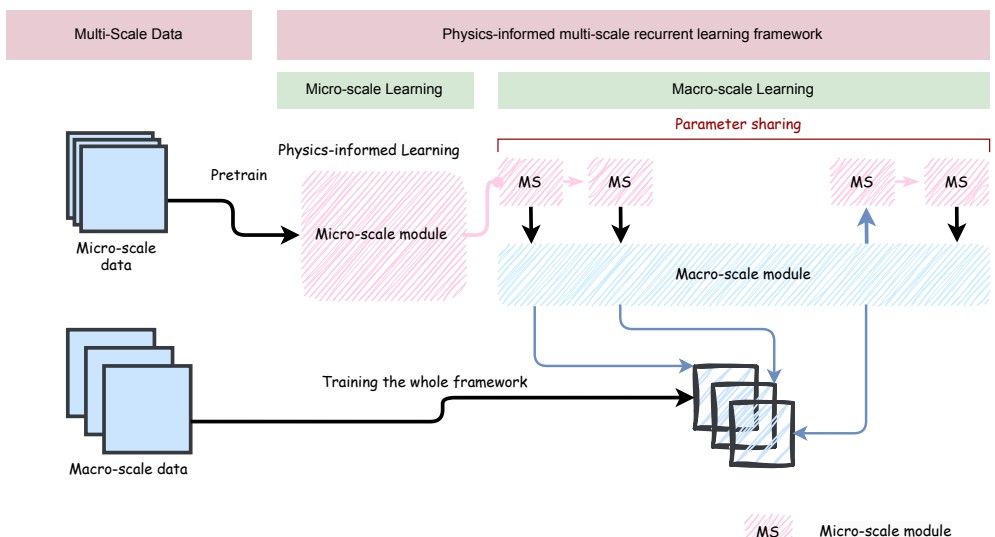

Figure S2: Overview of the PIMRL framework, which includes a pretraining stage using micro-scale data for physics-informed Learning and the utilization of a micro-module, informed by learned physics knowledge, to correct the macro-module during training on macro-scale data.

output of ConvLSTM into the physical space as residuals. The primary goal of the autoencoder in this context is to map an input to a low-dimensional latent space, and subsequently decode it to the original dimension at the output, facilitating the feature extraction and residual projection processes in the framework.

## C.3 RESIDUALLSTM

The structure of ResidualLSTM has been clearly illustrated in the main text. Here, we will elucidate the architecture of the ConvLSTMcell as the Figure S3.

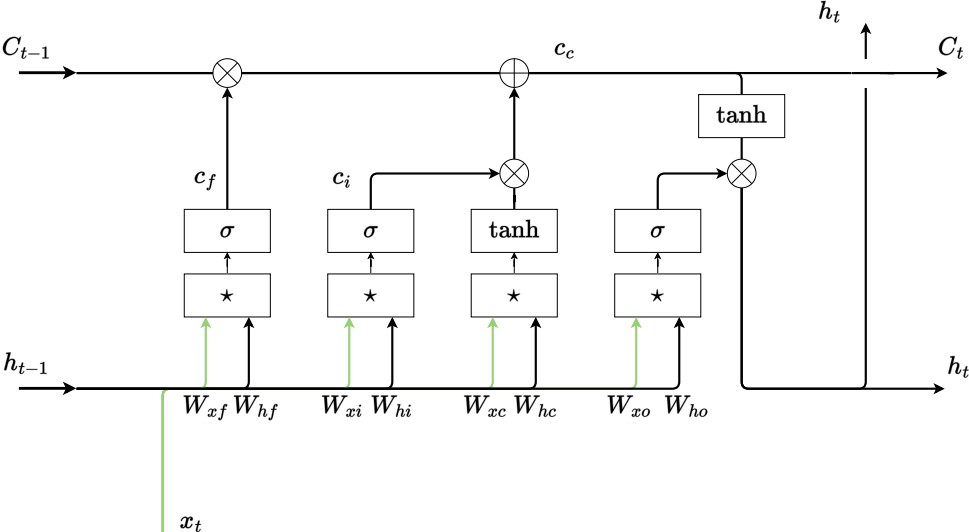

Figure S3: ConvLSTMCell

$$
\begin{aligned}
c_i &= \sigma(\mathrm{W}_{xi}(x_t) + \mathrm{W}_{hi}(h_{t-1})) \\
c_f &= \sigma(\mathrm{W}_{xf}(x_t) + \mathrm{W}_{hf}(h_{t-1})) \\
c_c &= c_f \cdot c + c_i \cdot \tanh(\mathrm{W}_{xc}(x_t) + \mathrm{W}_{hc}(h_{t-1})) \\
C_t &= \sigma(\mathrm{W}_{xo}(x_t) + \mathrm{W}_{ho}(h_{t-1})) \\
h_t &= c_o \cdot \tanh(c_c)
\end{aligned}
\tag{S2}
$$

The set of equations S2 presented above outlines the operations of a ConvLSTM cell. The equations involve computations of various gates and states within the cell, including input gate $c_i$, forget gate $c_f$, cell state $c_c$, output $C_t$, and the updated hidden state $h_t$. These equations govern the flow of information and transformations within the ConvLSTM cell, enabling the model to process spatiotemporal data efficiently by considering spatial dimensions in the calculations. The parameters $\mathrm{W}xi$, $\mathrm{W}hi$, $\mathrm{W}xf$, $\mathrm{W}hf$, $\mathrm{W}xo$, and $\mathrm{W}ho$, depicted in Figure S3, manage information from inputs $(x_t)$ and history $(h_{t-1}, C_{t-1})$ in a convolutional manner.

## D   BASELINE MODELS

In order to compare and evaluate the performance of our proposed method, we have trained multiple state-of-the-art (SOTA) baseline models as well as classical models, and compared them with our model. The introductions to each baseline model are presented below, while the training details are outlined in subsequent sections E.

**Fourier Neural Operator (FNO).** FNOLi et al. (2020) is a method that combines Fourier transforms with neural networks. This approach comprises two main components. The first component involves performing Fourier transforms on the system state quantities, learning certain information in the frequency domain, and then applying an inverse transform. The second component utilizes convolutions to process the system state quantities, complementing the information not captured during the frequency domain learning. The combination of these two components serves as the final result. We make two FNO models sharing the same architecture, which training in micro and macro scale datasets respectively and inferring in micro and macro time intervals. In the same model, we conducted training separately for two types of data, resulting in FNO and FNO-coarse.

**ConvLSTM.** ConvLSTM (Shi et al., 2015) is a specialized neural network architecture that combines convolutional and LSTM layers to effectively model spatial and temporal dependencies in sequential data.

**PeRCNN.** PeRCNN (Rao et al., 2023) represents a physics-informed learning methodology, embedding physical laws directly into the neural network architecture. It incorporates multiple parallel convolutional neural networks (CNNs), leveraging the simulation of polynomial equations through feature map multiplication. By doing so, PeRCNN augments the model's extrapolation and generalization capabilities.

## E   TRAINING DETAILS

All experiments were conducted on a single 80GB Nvidia A100 GPU, using an Intel(R) Xeon(R) Platinum 8380 CPU (2.30GHz, 64 cores). We only give some of the changed parameters here, and the other hyperparameters remain the same as the original text.

**PIMRL.** The architecture of the PIMRL model, illustrated in Figure 3, utilizes the Adam optimizer with a learning rate of $5 \times 10^{-3}$. The model have different parameters in different cases. More details were given by Table S2.

Additionally, we implement the StepLR scheduler to adjust the learning rate by a factor of 0.98 every 200 epochs. The pretraining details is same to the baseline model PeRCNN.

**FNO and FNO-coarse.** The network structure of FNO remains largely in line with the original study, with the primary adjustment being the adoption of an autoregressive training method for this model. We employ the Adam optimizer with a learning rate of $1 \times 10^{-3}$. More details are shown in the Table S8 and Table S9.

**ConvLSTM.** For ConvLSTM, we implement the ConvLSTM architecture like the macro-scale module of PIMRL. The StepLR scheduler is utilized with a step size of 200 and a gamma value of 0.98. The optimizer of choice is Adam, featuring a learning rate set at $1 \times 10^{-3}$. More details are shown in the Table S10.

**PeRCNN.** For PeRCNN, the training is different from the original paper since the micro-scale data only get a few pairs of continuous data. It is impossible to train PeRCNN in 400 to 800 steps at the same time. The details were shown in Table S5. We employ the Adam optimizer with learning rate of $1 \times 10^{-3}$ and the StepLR scheduler to adjust the learning rate by a factor of 0.98 per 200 epochs.

## F EVALUATION METRICS

In this paper, we have adopted some classical evaluation metrics such as RMSE, MAE and HCT. Root Mean Square Error (RMSE) quantifies the average error magnitude between estimated and actual values, serving as a gauge of the model's precision. Conversely, Mean Absolute Error (MAE) assesses the average absolute disparity between anticipated and observed values, denoting the true scale of discrepancies.

The definitions of these metrics are as follows:

$$\text{RMSE (Root Mean Square Error):} \quad \sqrt{\frac{1}{n} \sum_{i=1}^{n} (y_i - \hat{y}_i)^2}$$

$$\text{MAE (Mean Absolute Error):} \quad \frac{1}{n} \sum_{i=1}^{n} |y_i - \hat{y}_i| \tag{S3}$$

$$\text{HCT (High Correction Time):} \quad \sum_{i=1}^{N} \Delta t \cdot 1(\text{PCC}(y_i, \tilde{y}_i) > 0.8)$$

In the above equationsS3, $n$ represents the number of trajectories, $y_i$ represents the true value, and $\hat{y}_i$ represents the predicted value of the model. The PCC is the Pearson correlation coefficient, which is a statistical metric used to measure the linear correlation between two variables.

## G DATASET INFORMATIONS

The IC for the Kdv equation is created by summing multiple sine waves with random amplitudes, phases, and frequencies, resulting in a complex waveform. Initial conditions for the Burgers' equation are generated randomly according to a Gaussian distribution. The FN equation is initialized with random Gaussian noise for a warm-up period, after which time sequences are extracted to form the dataset. The GS equation starts the reaction from random initial positions then diffuses.

Table S1: Summary of experimental settings for different cases.(The 3D GS case is downsampled from $96^3$ to $48^3$ during training)

| Case | Numerical Methods | Spatial Grid | Time Grid | Training Trajectories | Test Trajectories |
|------|-------------------|--------------|-----------|----------------------|-------------------|
| Kdv | FVM | 256 | 0.01s | 5 | 2 |
| Burgers | FDM | $128^2$ | 0.001 | 13 | 3 |
| FN | FDM | $128^2$ | 0.5 | 5 | 3 |
| 2DGS | FDM | $128^2$ | 0.002 | 2 | 3 |
| 3DGS | FDM | $96^3$ | 0.25 | 3 | 2 |

**Korteweg-de Vries Equation.** The Korteweg-de Vries system, which elucidates the evolution of waves in nonlinear wave phenomena, can be described by the equation:

$$\frac{\partial u}{\partial t} = -u \frac{\partial u}{\partial x} - \frac{\partial^3 u}{\partial x^3} \tag{S4}$$

We got the 8 sets of data: 5 for training, 1 for validation, and 2 for testing. The data sets had spatial domain size $x \in [0, 64]$, where $\Delta t$ is 15 times $\delta t$ and $\delta t = 0.01s$.

**2D Burgers Equation.** The 2D Burgers' equation is commonly employed as a benchmark model for comparing and evaluating different computational algorithms, and describes the complex interaction between nonlinear convection and diffusion processes in the way like:

$$\frac{\partial u}{\partial t} = -uu_x - vu_y + \nu(u_{xx} + u_{yy}), \tag{S5}$$

$$\frac{\partial v}{\partial t} = -uv_x - vv_y + \nu(v_{xx} + v_{yy}). \tag{S6}$$

The $u_t$ and $v_t$ is the fluid velocities and $\nu$ denotes the viscosity coefficient. In this case, we choose $\nu = 0.005$ and the spatial domain size $x \in [0, 1]$, where $\Delta t$ is 8 times $\delta t$ and $\delta t = 0.001s$.

**2D FitzHugh-Nagumo Equation.** The FitzHugh-Nagumo system can be described by the equation:

$$\frac{\partial u}{\partial t} = \mu_u \Delta u + u - u^3 - v + \alpha, \tag{S7}$$

$$\frac{\partial v}{\partial t} = \mu_v \Delta v + (u - v)\beta. \tag{S8}$$

The coefficients $\alpha = 0.01$ and $\beta = 0.25$, governing the reaction process, take distinct values, while the diffusion coefficients are $\mu_u = 1$ and $\mu_v = 100$. In terms of time, $\Delta t = 15\delta t$ and $\delta t = 0.002s$.

**2D and 3D Gray-Scott Equation.** The Gray-Scott equations describe the temporal and spatial variations of chemical concentrations in reaction-diffusion systems, which can be described by the equation:

$$\frac{\partial u}{\partial t} = D_u \nabla^2 u - uv^2 + F(1 - u), \tag{S9}$$

$$\frac{\partial v}{\partial t} = D_v \nabla^2 v + uv^2 - (F + k)v. \tag{S10}$$

Here, in the two-dimensional case, $D_u$ and $D_v$ represent the diffusion coefficients of the two substances, with specific values of $Du = 2.0 \times 10^{-5}$ and $Dv = 5.0 \times 10^{-6}$. $F = 0.04$ denotes the growth rate of the substance, while $k = 0.06$ signifies its decay rate. In the 2D Gray-Scott case, we got 5 trajectories for training, 1 trajectory for validation and 3 trajectories for testing, where $\Delta t = 15\delta t$ and $\delta t = 0.5s$. In the three-dimensional case, we have the parameters: $DA = 0.2$, $DB = 0.1$, $F = 0.025$, and $k = 0.055$. We got 3 trajectories for training, 1 trajectory for validation and 2 trajectories for testing, where $\Delta t = 15\delta t$ and $\delta t = 0.25s$.

Table S2: Training Details of PIMRL.

| Case | Batchsize | Num of epochs |
|------|-----------|---------------|
| KdV | 512(all) | 5000 |
| Burgers | 8 | 5000 |
| 2DGS | 4 | 5000 |
| FN | 32 | 8000 |
| 3DGS | 16 | 8000 |

# H   SUPPLEMENT RESULTS

Table S3: The Result of U-NO, F-FNO, MWT and FNO in the FN Case.

| Metrics | U-NO | F-FNO | MWT | FNO |
|---|---|---|---|---|
| RMSE | 0.3675 | 0.2280 | 0.3494 | 0.1878 |
| MAE | 0.1465 | 0.1350 | 0.2228 | 0.1634 |

Table S4: Results with Error Bar under RMSE metric on all Cases.

| Model | PIMRL | PeRCNN |
|---|---|---|
| Kdv | $0.0457 \pm 0.0053$ | $0.0942 \pm 0.0082$ |
| Burgers | $0.0068 \pm 0.0006$ | $0.0075 \pm 0.0008$ |
| 2DGS | $0.0133 \pm 2.4 \times 10^{-12}$ | $0.0455 \pm 1.9 \times 10^{-11}$ |
| FN | $0.1349 \pm 0.0040$ | $0.1591 \pm 0.0061$ |
| 3DGS | $0.0381 \pm 0.0015$ | $0.0532 \pm 0.0027$ |

Table S5: Training Details of PeRCNN.

| Case | Batchsize | Num of epochs | Steps |
|---|---|---|---|
| KdV | 512(all) | 1000 | 45 |
| Burgers | 32 | 1000 | 16 |
| 2DGS | 32 | 1000 | 45 |
| FN | 36 | 1000 | 45 |
| 3DGS | 32 | 1000 | 45 |

Table S6: Comparison of RMSE and MAE for different cases.

| Case | RMSE | MAE |
|---|---|---|
| FN | 0.2803 | 0.2482 |
| 2DGS | NaN | NaN |

Table S7: Running Time, Parameter Size, and GPU Memory for PIMRL, U-NO, MWT, FNO, and ConvLSTM

| Model | Running Time | Parameter Size | GPU Memory |
|---|---|---|---|
| PIMRL | 9 s | 3.33 M | 1728 M |
| U-NO | 7 s | 15.29 M | 2320 M |
| MWT | 12 s | 0.09 M | 1732 M |
| FNO | 2 s | 8.39 M | 1580 M |
| ConvLSTM | 5 s | 3.32 M | 1708 M |

Table S8: Training Details of FNO.

| Case | Batchsize | Num of epochs | Steps |
|---|---|---|---|
| KdV | 512(all) | 1000 | 45 |
| Burgers | 32 | 2000 | 16 |
| 2DGS | 32 | 2000 | 45 |
| FN | 36 | 2000 | 45 |
| 3DGS | 32 | 2000 | 45 |

Table S9: Training Details of FNO-coarse.

| Case | Batchsize | Num of epochs |
|---|---|---|
| KdV | 512(all) | 5000 |
| Burgers | 32 | 5000 |
| 2DGS | 32 | 5000 |
| FN | 32 | 8000 |
| 3DGS | 16 | 8000 |

Table S10: Training Details of ConvLSTM.

| Case | Batchsize | Num of epochs |
|---|---|---|
| KdV | 512(all) | 5000 |
| Burgers | 8 | 5000 |
| 2DGS | 4 | 5000 |
| FN | 32 | 8000 |
| 3DGS | 16 | 8000 |

