# OpenReview forum: "PIMRL: Physics-Informed Multi-Scale Recurrent Learning for Spatiotemporal Prediction"
_ICLR.cc/2025/Conference — Submitted to ICLR 2025_

### Official Review · Reviewer_BVvK · 2024-10-27

**Soundness:** 3
**Presentation:** 2
**Contribution:** 2
**Rating:** 6
**Confidence:** 5

**Summary:**

This paper focuses on the temporal multi-scale property of spatiotemporal systems and proposes a multiscale recurrent learning framework, named PIMRL. In this model, the micro-scale module learns to embed multiscale data and the macro-scale module captures the data-driven evolution. PIMRL performs well in five different tasks.

After rebuttal: I think most of my concerns have been resolved. Thus, I have raised my score to 6 and changed the score of presentation and soundness.

**Strengths:**

This paper focuses on an important problem: the temporal multiscale property, which is mainly overlooked in previous methods.

The experiments are relatively comprehensive. The authors test PIMRL in five typical tasks.

**Weaknesses:**

1.	About the writing.

I believe that this paper needs a significant revision to ensure a clear presentation. Here are some examples.

(1)	Introduction: There are many important concepts without explanation. For instance, the authors claim that the micro-scale is “pretrained” in the abstract and introduction. However, in the method part, there are no descriptions of how this pretraining be implemented.

(2)	Related work: The authors fail to provide a clear organization of the related work. For example, ConvLSTM is just one classical (maybe old) baseline in spatiotemporal learning. There are extensive papers that weren’t included, such as PredRNN [1] or TrajGRU [2].

[1] PredRNN: a recurrent neural network for spatiotemporal predictive learning

[2] Deep Learning for Precipitation Nowcasting: A Benchmark and A New Model

(3)	Method: I think Eq. (2) and (3) do not provide a clear and right formalization for the overall framework. The descriptions of the prediction protocol are self-contradictory. Given that each micro module can conduct a \delta t stepsize transition and the macro module is \Delta t, why the output of Eq.(3) is just t+k\delta t. Besides, as they presented in Figure 2, I cannot tell which part is “history information” or if the micro-scale module is applied to the model prediction or not. A flowchart or pseudocode can be helpful.

The low quality of writing seriously affects the contribution of this paper.

2.	Compare with more advanced baselines.

As the authors mentioned in Lines 125-133, there are many advanced Neural Operators such as MWT and U-NO, which should be compared.

3.	About the novelty.

As the proposed method is more like a multiscale ensemble method, I cannot score high for the novelty.

**Questions:**

Why did the authors name the micro module as “physics-informed”? Actually, I think the formalization of Eq. (4) is just a convolution neural work, which is far away from the conventional definition of “physics-informed neural networks”.

---

> ### Author Response · Authors · 2024-11-22
> **Reply to reviewer BVvK (Part 1)**
>
> Thank you for your constructive comments. We understand that there might be some misunderstanding regarding our work. We would like to clarify them in detail as follows.
>
> **Weakness 1a: Low quality of writing.**
>
> **Reply:** Thanks for your feedback. Following your comment, we have thoroughly proofread our paper, corrected typos and grammar mistakes, and re-organized our writing to improve the clarity of the paper. The majority of the paper has been re-written (marked in red color). We believe the presentation has been substantially improved. Please refer to the **updated .pdf file**.
>
> **Weakness 1b: Explanation of micro-scale module pretraining.**
>
> **Reply:** Great remark! We have now re-written the subsection of Training Strategies (see below and also Section 3.4, Page 4), where we have provided details on the training strategies for PIMRL (both miscro- and macro-scale modules). Additionally, we have included an ablation study with the model that omits pre-training to validate the effectiveness of our training approach (see Section 4.2, Page 9) in the revised paper.
>
> "*Since the PIMRL model consists of micro- and macro-scale modules, adopting brute-force end-to-end training yields unsatisfactory results (see the ablation study). Firstly, we establish a pre-training phase where only the micro-module is trained. The purpose of this pre-training phase is to enable the micro-module to effectively learn the dynamics of the physical system and the underlying physical laws, free from the influence of the macro-module. In the next phase, referred to as the overall training phase, all modules within the PIMRL framework are engaged in training together. During this phase, the micro-module benefits from the parameters pre-trained in the pre-training phase, serving to supervise and correct the macro-module. The output from the macro module serves as the final output of the entire PIMRL framework and is used to compute the loss. The details are shown in Appendix C.1.*"
>
> **Weakness 1c: Improve organization of the related work.**
>
> **Reply:** Excellent suggestion! We have thoroughly re-written the Related Work section and added the PredRNN [1] or TrajGRU [2] (please see Section 2, Page 3) revised paper.
>
> ***References:***
>
> [1] Wang Y et al., PredRNN: a recurrent neural network for spatiotemporal predictive learning. NeurIPS, 2017.
>
> [2] Shi X et al., Deep Learning for Precipitation Nowcasting: A Benchmark and A New Model. NeurIPS, 2017.
>
> **Weakness 1d: Improve the formalization for the overall framework.**
>
> **Reply:** The formalization for the overall framework (e.g., the inference workflow) is described as follows:
>
> - Firstly, the micro-module loop is a simple autoregressive process with time step $\delta t$, where the output at the previous time step serves as the input for the next time step.
> - Secondly, the macro-module loop performs self-cycles. When the micro-module is involved in the prediction, for every $k$ steps of micro-module with $\delta t$, the final output of micro-module is passed to the macro-module, and at this point, the output from the macro-module serves as the output of the entire PIMRL model. When the micro-module is not involved in the prediction, the macro-module loop is a simple autoregressive process with time step $\Delta t$ shown as Equation **Macro-module Loop**.
> - Finally, there is a PIMRL loop that operates in conjunction with the macro-module loop. After every $N$ cycles of the macro-module loops, the micro-module stops participating in the prediction, and the macro-module performs $N$ steps of autoregressive prediction. This completes a total of $2N$ cycles. Each output from the macro-module during these $2N$ cycles serves as the output of PIMRL as depicted in Equation **PIMRL Loop**.
>
> The formulations of the solution update can be expressed as:
> $$\text{PIMRL Loop: } \mathbf{u} _{t+2N\Delta t }   = \underbrace{F _{macro}(...F _{macro}} _{\times N} (\mathbf{u} _{t+Nk\delta t})),$$
>
> $$\text{Macro-module Loop: }\mathbf{u} _{t+2k\delta t }   = F _{macro}( \underbrace{F _{micro}(...F _{micro}} _{\times k} (\mathbf{u}_t)) ).$$
>
> We have re-written the Network Architecture subsection (Section 3.3, Page 4) and included the above contents in revised paper.
>
> **Weakness 2: Compare with more advanced baselines.**
>
> **Reply:** Great suggestion! We took your advice and performed tests on the FN dataset for three additional baseline models (e.g., **U-NO**, **F-FNO** and **MWT**). The experimental results for these new models tested on the coarse-grid data are reported in **Table A** below, which have also been provided as Table S3 (Appendix H, Page 17) in the revised paper.
>
> **Table A: The results of U-NO, F-FNO, MWT and FNO in comparison with PIMRL.**
> |Metrics|U-NO|F-FNO|MWT|FNO|PIMRL|
> |-|-|-|-|-|-|
> |RMSE|0.3675|0.2280|0.3494|0.1878|**0.1349**|
> |MAE|0.1465|0.1350|0.2228|0.1634|**0.0990**|

---

> ### Author Response · Authors · 2024-11-22
> **Reply to reviewer BVvK (Part 2)**
>
> **Weakness 3: Clarify the Novelty.**
>
> **Reply:** We would like to clarify that the PIMRL is *not* a simple ensemble method. As we know, existing nerual methods face critical challenges of inevitable error accumulation in long-term prediction. We develop a unique multi-scale learning architecture to tackle this issue, which well controls error accumulation, with the novel contributions described as follows:
>
> - We proposed a new PIMRL framework that effectively leverages information from multi-scale data for long-term spatiotemporal dynamics prediction. The concept of reducing the accumulation of errors is achieved through the integration of the macro-scale and micro-scale modules.
> - We designed a novel message passing mechanism between micro- and macro-scale modules (through hybrid parallel and serial connections), which effectively transmits physical information, enhances the micro-module's correction capability, and reduces the number of micro-scale rollout iterations through the macro-scale module.
> - The PIMRL model achieved optimal performance in effectively predicting the duration of multiple different cases from fluid dynamics to physical systems, demonstrating its scalability and laying a solid foundation for more generalizable models in the future.
>
> In particular, our model distinctly differs from simple serial or parallel ensemble methods. Such a new model architecture can effectively deal with *multi-scale Burst-sampled data* for long-term prediction of spatiotemporal dynamics. Hope this clarifies your concern about our novelty.
>
> **Question 1: Explanation of "Physics-Informed".**
>
> **Reply:** Thank you for your question. When a certain term in the governing PDEs remains known (e.g., the diffusion term $\Delta \mathbf{u}$), its discretization can be directly embedded in PeRCNN (called the physics-based Conv layer). The convolutional kernel in such a layer can be set according to the corresponding finite difference (FD) stencil. In essence, the physics-based Conv connection is constructed to incorporate known physical principles, whereas the $\Pi$-block is aimed at capturing the complementary unknown dynamics (Section 3.5, Page 5). We believe this process can be considered as "physics-informed".
>
> **Concluding remark:** Once again, thank you for your comments and suggestions. We hope the above responses are helpful to clarify your concerns. Looking forward to your feedback!

---

> ### Author Response · Authors · 2024-11-23
> **Looking forward to your feedback**
>
> Dear Reviewer BVvK,
>
> Again, thanks for your constructive comments. We would like to follow up on our rebuttal to ensure that all concerns have been adequately addressed. If there are any further questions or points that need discussion, we will be happy to address them. Your feedback is invaluable in helping us improve our work, and we eagerly await your response.
>
> Thank you very much for your time and consideration.
>
> Best regards,
>
> The Authors

---

> ### Comment · Reviewer_BVvK · 2024-11-24
>
> I would like to thank the authors' response, which improves the overall presentation quality of this paper. Also, thanks for the additional experiments. However, there are still some serious issues unsolved.
>
> >  I have to say, that the current presentation is still significantly below the bar of ICLR papers.
>
> **(1) There are some unexplained symbols.** For example, in Figure 1, I cannot find any descriptions about $\delta_\tau$ or $\zeta_t$. In Figure 2, $u_0$, $u_{k \delta t}^{micro}$ or even $\hat{u}$ are without any clear statements. After reading the current method part for around 1 hour, I can understand what the authors mean. Thus, I strongly suggest they rephrase Equations 2 and 3. Specifically, omitting the subscript, the output of equation 2 should be $\hat{u}$ and equation3 should be split into two separate parts, where one's output is $u_{k \delta t}^{micro}$ and the other part is $\hat{u}$.
>
> **(2) Not smooth logic in Section 3.3.** Even if the authors have revised these foundational writing issues, the current method part is still hard to read. I suggest they split Figure 2 into 2 figures, where one is for the overall framework (I do not think Section 3.3 is for "network architecture", it is a "forecasting framework") and another is for detailed module design.
>
> **(3) About related work.** Firstly, thanks for the authors' effort in including related works. Although highlighting this seems too naive, I have to say they should add some discussion about the relation between their method and related works. Without this, I cannot place the PIMRL in the right position in this community.
>
> Again, as an academic paper, the writing quality can highly impact the knowledge spread. Thus, I think the authors should keep revising this paper.
>
> > About the forecasting formalization in Figure 2.
>
> I can understand that $F_{micro}$ can deliver a temporal transition in $\delta t$ and the $F_{macro}$ is for a temporal transition of $\Delta t$. However, this will make the third column for Figure 2(a) incorrect, where the input is $Nk\delta t$ and the output is $N\Delta t$.
>
> In addition, I think all the loss functions should be calculated in the $F_{macro}$'s output. Why the loss function is applied to the input in the last column of Figure 2(a).
>
> Further, if the authors try to deliver some information by the underlying color. Why the right-bottom part is colored pink?
>
> Also, the usage of $\pi$-block is also misleading, since it is also used for the "Elementwise product" operation.
>
> > About the motivation of PIMRL.
>
> I think the authors fail to clarify why PIMRL can reduce the long-term accumulation error. As they stated in lines 83-85 of the revised paper, the micro is used for learning "physics" and the macro is to reduce rollout error. However, the micro-module requires many rollouts. How do the authors tackle the rollout accumulation error caused by the micro-scale module?
>
> Besides, since PIMRL combines two different models, how about the efficiency (running time, parameter size, GPU memory) of PIMRL w.r.t. other deep learning baselines (e.g. U-NO, MWT, FNO, ConvLSTM)?
>
> > About the so-called "physics-informed" micro-scale module.
>
> Firstly, I agree with the authors that convolution neural networks can represent finite difference methods. However, I think, in this paper, the motivation for using finite difference methods is self-contradictory. As they stated in Lines 37-38, DNS necessitates high spatial resolution and fine time stepping. If the micro-scale module is trying to approximate a finite difference, how can it deal with the low-resolution inputs? That is why I mentioned PINNs in my original review. PINN adopts the auto-differential operations in deep learning frameworks (e.g. Pytorch and JAX), which can ensure an accurate approximation.
>
> > About experiments.
>
> I appreciate that the authors mentioned many recent progress in PDE solving (Lines 124-129). Why some of them are compared and some not? I think they should include more comparisons of advanced neural operators in the revised paper.
>
> In conclusion, I think the idea of incorporating slow and fast modules in prediction is reasonable. But I cannot rank high for this idea, since the design of the micro-scale module and macro-scale module is more like a trade-off, where the former is for accurate modeling of temporal differential and the latter is for reducing accumulation error. I do not think this paper really tackles the temporal modeling problem in spatiotemporal modeling.
>
> Thus, I raise my score of contribution to fair and my overall score to 3 with confidence 5.

---

> > ### Author Response · Authors · 2024-11-25
> > **Reply to the further comments from Reviewer BVvK (Part 2)**
> >
> > >**Weakness 3: About the motivation of PIMRL.**
> >
> >  **Reply:** While the micro-module in PIMRL does introduce errors, **the degree of error accumulation is not significant** due to the **much fewer** iterations compared to the PeRCNN model. By reducing the number of micro-module iterations, PIMRL can predict long-term physical states before the accumulated errors become unacceptable. Furthermore, the transmission mechanism in PIMRL allows PeRCNN to convey its learned physical information even when it is not involved in every prediction step, thereby correcting the outputs of our macro-module. Thanks to the design of the PIMRL framework, the cumulative error in PIMRL is significantly smaller compared to previous methods.
> >
> > Although PIMRL combines two different modules, its efficiency is still comparable to the baseline models, as shown in **Table A** below, which have also been provided as **Table S7 (Appendix H, Page 18)** in the revised paper.
> >
> > **Table A: Running Time, Parameter Size, and GPU Memory for PIMRL, U-NO, MWT, FNO, and ConvLSTM.**
> >
> > | Model      | Running Time | Parameter Size | GPU Memory |
> > |------------|--------------|----------------|------------|
> > | **PIMRL**      |  9 s         | 3.33 M         | 1728 M     |
> > | U-NO       | 7 s          | 15.29 M        | 2320 M     |
> > | MWT        | 12 s         | 0.09 M         | 1732 M     |
> > | FNO        | 2 s          | 8.39 M         | 1580 M     |
> > | ConvLSTM   | 5 s          | 3.32 M         | 1708 M     |
> >
> >
> > >**Weakness 4: Clarify micro-scale module.**
> >
> > **Reply:** Insightful comment! Firstly, in all the problems we are solving in this work, we assume that the explicit formulation of governing PDEs are unknown *a priori* knowledge. Hence, PINN methods are **not** applicable. Secondly, the unique $\Pi$-block is designed to approxiamte the underlying PDEs given very limited training data and to produce solutions even on relatively coarse mesh grids compared with DNS methods such as finite difference (e.g., 48 $\times$ 48 $\times$ 48 for the 3D GS case). Given the same amount of training data (miscro time scale), we can see from **Table B** (below) that PeRCNN significantly outperforms other baseline models for all the datasets. The baseline models fail to make rational predictions at the micro scale because of very limited training data used herein (only a few Burst-sampled trajectories), while PeRCNN performs well because it correctly learnable the underlying dynamics of the physical system. Therefore, we conclude that PeRCNN, pre-trained on micro-scale data, could well serve as the micro-scale simulator.
> >
> >
> > **Table B: Test on different choices of micro-scale modules.**
> >
> > |Metrics|FNO| |PeRCNN| |
> > |-|-|-|-|-|
> > |Case|RMSE|MAE|RMSE|MAE|
> > |KdV (Grid: 256)|0.4891|0.3300|**0.0942**|**0.0942**|
> > |2D Burgers (Grid: 128 $\times$ 128)|0.1561|0.1301|**0.0075**|**0.0058**|
> > |2D FN (Grid: 128$\times$128)|937|$>10^3$|**0.1591**|**0.1139**|
> > |2D GS (Grid: 128 $\times$ 128)|NaN|NaN|**0.0455**|**0.0268**|
> > |3D GS (Grid: 48 $\times$ 48 $\times$ 48)|0.2798|0.1950|**0.0532**|**0.0977**|
> >
> >
> > >**Weakness 5: About experiments where more advanced neural operators should be compared.**
> >
> > **Reply:** Thanks for your suggestion. However, we would like to clarify that we have conducted extensive experiments on baseline comparison, including representative methods such as ConvLSTM, PeRCNN, FNO, U-NO, F-FNO, and MWT. It is important to note that our tasks provide only **very limited training data**, making it **challenging** for **exisiting neural operators** to effectively learn the underlying dynamics. Additionally, we aim to develop a model for **long-term predictions** spanning hundreds of macro-scale time steps, which is highly challenging. We would like to draw your attention that the majority of neural operators in the literature are trained againt rich datasets (e.g., $\sim 1000$ or more trajectories for dynamics problems). Given, the very small amount of data (e.g., only 2 trajectories with a few thousand sampled data points for the FN case), it is not surprising to see the baseline models fail. While testing more neural operators as baselines would add more contents to the results, it will be redundant in the regard mentioned above. Hope this clarifies your concern.

---

> > ### Author Response · Authors · 2024-11-25
> > **Reply to the further comments from Reviewer BVvK (Part 3)**
> >
> > > "I think the idea of incorporating slow and fast modules in prediction is reasonable. But I cannot rank high for this idea, since the design of the micro-scale module and macro-scale module is more like a trade-off, where the former is for accurate modeling of temporal differential and the latter is for reducing accumulation error. I do not think this paper really tackles the temporal modeling problem in spatiotemporal modeling."
> >
> > **Reply:** We appreciate your comment, but respectfully **disagree** on it. The architecture that unifies the micro-scale and macro-scale modules is **not** a simple trade-off. In fact, there is no so-called "trade-off" in the network design. The unique design of the network achitecture is rooted in a novel message passing mechanism between micro- and macro-scale modules (through hybrid parallel and serial connections), which effectively transmits physical information, enhances the micro-module's correction capability, and reduces the number of micro-scale rollout iterations through the macro-scale module. In so doing, the PIMRL framework effectively leverages information from multi-scale data for long-term spatiotemporal dynamics prediction. Reducing the accumulation of errors is achieved through the integration of the macro-scale and micro-scale modules. As a result, our model successfully tackles the most fundamental challenge in spatiotemporal modeling, aka, **long-term prediction under the environment of very limited training data**. This has been fully evidenced by our model's superior performance and extensitive comparison with the baseline models.
> >
> > Hence, we feel confused about the reviewer's comment "*I do not think this paper really tackles the temporal modeling problem in spatiotemporal modeling.*" We really appreciate if you could elaborate a bit more about that.
> >
> > **Concluding remark:** Once again, thank you for your comments and suggestions. We very much look forward to your feedback!

---

> > > ### Author Response · Authors · 2024-11-26
> > > **Reply to the further comments from Reviewer BVvK (Part 4)**
> > >
> > > **In addition to the primary experiments**, we conducted an extended study to further investigate the role of individual components within the PIMRL framework. Specifically, this additional experiment involved the **removal of the macro-module's autoregressive component from the latter part of the PIMRL framework**. As a result, the modified model relies solely on the simultaneous operation of micro- and macro-modules for prediction tasks, thereby **eliminating the original strategy for controlling error accumulation**. We designate this revised model as the **"Non Accumulative Error Control" (NAEC)** model. The comparative results between the NAEC model and the original PIMRL framework are presented in **Table C**. Through this comparative analysis, our study aims to provide deeper insights into the specific impact of the error accumulation control mechanism on overall model performance.
> > >
> > > **Table C: The Results of **NAEC** and PIMRL in the FN Case.**
> > >
> > > |Metrics|NAEC|PIMRL|
> > > |-|-|-|
> > > |RMSE|0.1581|**0.1349**|
> > > |MAE|0.1514|**0.0990**|
> > >
> > > The experimental results presented in **Table C** demonstrate that controlling error accumulation significantly outperforms the combined predictions of micro- and macro-modules. Guided by this direct and fundamental approach, the simplified version of PIMRL achieves superior performance in this experiment. This outcome underscores the critical importance of the method for controlling error accumulation proposed in our paper, **highlighting its effectiveness and potential value in enhancing model performance**.

---

> > ### Author Response · Authors · 2024-11-27
> > **Looking forward to your feedback on our reply to your further comments**
> >
> > Dear Reviewer BVvK,
> >
> > Again, thanks for your constructive comments, which are very much helpful for improving our paper. Our responses to your additional comments have been posted. If there are any further questions or points that need discussion, we will be happy to address them. Your feedback is invaluable in helping us improve our work, and we eagerly await your feedback.
> >
> > Moreover, we have further revised our paper based on your additional comments and suggestions. Please refer to **the updated .pdf file** (marked in blue color).
> >
> > Your consideration of updating your score will be much appreciated! Thank you.
> >
> > Best regards,
> >
> > The Authors

---

> ### Author Response · Authors · 2024-11-25
> **Reply to the further comments from Reviewer BVvK (Part 1)**
>
> We sincerely thank you for the additional feedback. The following responses have been incorporated **(indicated in blue color)** into the revised paper (please see ***the updated .pdf file***).
>
> >**Weakness 1.1: Explanation of unexplained symbols.**
>
> **Reply:** Thanks for great suggestion! The following explanation has been provided in the Introduction (Section 1, Page 2) and Methodology sections (Section 3, Page 4-5) in the revised paper.
>
> **1. Explanation of $\Delta \tau$ and $\zeta_ t$:** In the multi-scale sampling, where $\Delta \tau$ denotes the micro-scale time interval for fast dynamics, $\Delta t$ the macro-scale time interval for slow dynamics, and $\zeta_ t$ the scale separation variable (typically $\zeta_ t < 1$ or $\zeta_ t \ll 1$). (Section 1, Page 2, Figure 1)
>
> **2. Explanations of $\mathbf{u}_ 0$, $\mathbf{u}^{micro}_ {k\delta t}$ and $\hat{\mathbf{u}}$:** (Section 3, Page 4, Figure 2)
> - $\mathbf{u}_ 0$: Initial state of the system.
> - $\mathbf{u}^{micro}_ {k\delta t}$: State predicted by the micro-module after $k$ iterations.
> - $\hat{\mathbf{u}}$: Predicted value **from PIMRL** of the physical state.
>
> **3. Rephrased Equations:**(Section 3.3, Page 5)
>
> $$\text{PIMRL Loop: } \mathbf{\hat{u}}_ {t+2N\Delta t } =\underbrace{F_ {macro}(...F_ {macro}}_ {\times N} (\mathbf{\hat{u}}_ {t+Nk\delta t})),$$
>
> $$\text{Micro-module Loop: } \mathbf{u}^{micro}_ {k\delta t} = \underbrace{F_ {micro}(...F_ {micro}}_ {\times k} (\mathbf{u}_ t)),$$
>
> $$\text{Macro-module Loop: }\mathbf{\hat{u}}_ {t+2k\delta t }  = F_ {macro}( \mathbf{u}^{micro}_ {k\delta t} ).$$
>
> >**Weakness 1.2: Enhancing Readability of Section 3.3 and Refining Figure Presentation.**
>
> **Reply:** Following your suggestion, we have split **Figure 2** into two figures **Figure 2** and **Figure 3** in the revised paper (Section 3 on pages 4-5), and we have changed the title of Section 3.3 to '**Forecasting Framework**'.
>
> **Weakness 1.3: The Relationship between PIMRL and Related Work.**
>
> **Reply:** To address the high computational cost and slow speed associated with traditional numerical methods, PIMRL adopts deep learning techniques. However, conventional deep learning approaches often fail to fully leverage physical information. PeRCNN demonstrates that incorporating physical methods can be effective; therefore, we opt to utilize this physics-embedded approach. To mitigate error accumulation and fully exploit multi-scale traning data, we employ a multi-scale framework. Given that the coupling strategies used in some hybrid methods are inadequate, we propose a novel coupling method within PIMRL to effectively transmit embedded physical information. We have added connections mentioned above to Related Work (please see Section 2, Page 3).
>
> >**Weakness 2: About the forecasting formalization in Figure 2.**
>
>  **Reply:** Excellent comment! Here are our revisions as follows.
>
>  - We have rewritten the equations and adjusted Figure 2. Specifically, we have refined the third column to represent $(N-1)k\delta t$.
>
>  - All loss functions are calculated at the output of the macro-modules.
>
>  - In the bottom-right section, which is now shaded gray (previously pink), it indicates that the PIMRL loop restarts, and the loss is computed based on the output of the macro-module. The use of color in this context serves to highlight that the output of the macro-module serves dual purposes: it is both the input to the micro-module **(below the macro-module, shaded gray)** and the output **(above the macro-module)** used for computing the loss in PIMRL.
>
>  - Additionally, the **$\Pi$-block** in Figure 3 denotes the **"Elementwise Product"** operation (**not the micro-module anymore**), which is a component of the micro-module.

---

> ### Comment · Reviewer_BVvK · 2024-11-29
>
> Thanks for your response and effort in revising the paper. Your experiments about model efficiency comparison and ablations of Accumulative Error Control and micro-scale modules. are received. However, my key concerns about presentation issues and motivation still exist.
>
> > Again and again about writing issues.
>
> Note that I am not talking about some meaningless typos. I want to point out that this paper has some foundational issues ranging from the original revision, the first rebuttal, and even the current version, which impends the possible readers to understand the author's design. That is why I rate this paper as 1 for the original version.
>
> For the first two versions, you can find the details from my previous review. Although I thought I understood your first rebuttal version, the current version confused me again. Here are the details about the current version.
>
> - Equation 3 should be Micro-
>
> - Lines 231-234, under what context, the micro-module is or is not involved in the prediction?
>
> - Lines 235-239, I am totally confused about "loop" and "cycles", can you use more distinguishable words for these two concepts? Also, which part in Figure 2 is "cycle"?  I cannot see any other cycles in this figure except the mirco-module loop and macro-module loop. Also, from Figure 2, which part is the past observation and which part is the future prediction?
>
> - "Further, if the authors try to deliver some information by the underlying color. Why the right-bottom part is colored gray?"
>
> Can you give me a detailed example, such as predicting the future 10 steps based on the past 10 observations? I have spent several hours on page 5 but still cannot understand the forecasting framework.
>
> Further, there are some unsupported and self-contradictory designs. "After every N−1 cycle of the macro-module loops, the micro-module stops participating in the prediction". Why does the micro-module only participate in the first N−1 cycle and but do not attend the next N steps? According to your description, the forecasting framework is "autoregressive", does this mean Equations 2-4 will be applied for every new forecasting step? I cannot tell autoregressive from current formalization.
>
> Thus, I keep my opinion about this paper's writing quality.
>
> > Wrong position about previous methods.
>
> “It is important to note that our tasks provide only very limited training data, making it challenging for existing neural operators to effectively learn the underlying dynamics.” I cannot tell what is the difference between PIRML and other methods. All the baselines are purely data-driven, why PIRML is better in limited training data.
>
> > I do not think this paper really tackles the temporal modeling problem in spatiotemporal modeling.
>
> This paper highlights reducing accumulation error as their contribution, I do not think the current design can really reduce accumulation error from a foundation perspective. For example, the autoregressive steps in PIRML can be larger than other models such as FNO or MWT. If one method takes more autoregressive steps, how can we say it is better in controlling accumulation error? In most cases, the accumulation error can only be controlled by multistep rollout. Can the authors provide some theoretical analysis for this contribution?
>
> In conclusion, I will keep the score and be active in the next reviewer-AC discussion period. I also want to hear about other reviewers' opinions about this paper's writing and novelty.

---

> > ### Author Response · Authors · 2024-12-01
> > **Clarification of your misunderstandings (part 2/3)**
> >
> > >**Comment 4.** Lines 235-239, I am totally confused about "loop" and "cycles", can you use more distinguishable words for these two concepts? Also, which part in Figure 2 is "cycle"? I cannot see any other cycles in this figure except the mirco-module loop and macro-module loop. Also, from Figure 2, which part is the past observation and which part is the future prediction?
> >
> > **Reply:** We would like to clarify that "loop" and "cycle" are equivalent (please see our reply to **Comment 1**). We define a complete loop of PIMRL as a basic temporal block which includes 8 rollouts of the macro-module and 45 rollouts of the micro-module. We will use "loop" and "rollout" in the revised paper to avoid any misunderstanding.
> >
> > Based on your comment "*which part is the past observation and which part is the future prediction*", we confirmed that you might have confused our model with typical time series forecasting models! This is a critical misunderstanding. We would like to emphasize again that *we aim to establish a surrogate model to predict spatiotemporal dynamics at a long-term horizon only given a single initial condition of the system*. **This is the fundamental basis to correctly evaluate our work**. Hence, in our framework, all the outputs are predictions at different time steps (including micro- and macro-scale modules) made by the PIMRL, except $\mathbf{u_0}$ which is an initial condition.
> >
> > >**Comment 5.** Further, if the authors try to deliver some information by the underlying color. Why the right-bottom part is colored gray (in Figure 2)?
> >
> > **Reply:** The gray areas in Figure 2 indicate the part where the micro-module of PIMRL performs predictions, highlighting their role as the micro-module rollouts at the beginning of each PIMRL loop.
> >
> > >**Comment 6.** Further, there are some unsupported and self-contradictory designs. "After every N−1 cycle of the macro-module loops, the micro-module stops participating in the prediction". Why does the micro-module only participate in the first N−1 cycle and but do not attend the next N steps? According to your description, the forecasting framework is "autoregressive", does this mean Equations 2-4 will be applied for every new forecasting step? I cannot tell autoregressive from current formalization.
> >
> > **Reply:** Firstly, we clarify that "autoregressive" means that the output at one timestamp serves as the input for the next timestamp. Specifically, the final output of the PIMRL loop serves as the input for the next PIMRL loop. Additionally, after the micro-module's participation ends, the output of the macro-module at each timestamp serves as the input for the next timestamp. For the micro-module, its input is the output from the previous timestamp, except at the beginning of the PIMRL loop, where it receives the initial input. Obviously, this is an autoregressive form.
> >
> > >**Comment 7.** I cannot tell what is the difference between PIRML and other methods. All the baselines are purely data-driven, why PIRML is better in limited training data.
> >
> > **Reply:** We feel confused about your comment. Firstly, the netwok architecture of our model (a multiscale framework) is fundamentally different from the existing baselines. Secondly, the superior model performance is due to the unique multiscale learning strategy which better controls the error propagation. We believe, if all your misunderstandings are cleared (please see our response to your above comments) and if you are familiar with the baseline models, you would feel straighforward to tell the difference.

---

> ### Author Response · Authors · 2024-12-01
> **Clarification of your misunderstandings (part 1/3)**
>
> After reading your additional comments, we found there exists a **significant misunderstanding** of our work. We clarify these misunderstandings as follows.
>
> On a separate note, we believe your critism on the motivation and writing of our paper is **completely biased**. Throughout your entire comments, we feel extremely confused and surprised you mentioned repeatedly that "*you spent several hours but failed to understand the overall framework*"! We wonder what motivated you to change your confidence score from 4 to 5, paticularly in the situation that our work was not well understood. Please clarify.
>
> Please note that, seriously, we found that many of your comments are rooted in **misunderstanding of our work**. We are not convinced by your very low score rating!
>
> Our responses to your additional comments are given as follows.
>
> >**Comment 1.** Provide a detailed example, such as predicting the future 10 steps based on the past 10 observations? I have spent several hours on page 5 but still cannot understand the forecasting framework.
>
> **Reply:** Let us use the FN example for illustration. We define a complete loop of PIMRL as a basic temporal block which includes 8 rollouts of the macro-module and 45 rollouts of the micro-module. In the FN case, the input to PIMRL consists only of the initial state $\mathbf{u_0}$. After receiving $\mathbf{u_0}$, PIMRL simultaneously inputs it into both the micro-module and the macro-module. The micro-module undergoes 15 rollouts before providing its output to the macro-module. Over the course of 45 rollouts, the micro-module outputs to the macro-module 3 times as **the second, the third and the fourth** inputs. After these 4 rollouts, the macro-module continues with autoregressive prediction, where its output at each timestamp serves as the input for the next timestamp, without further involvement from the micro-module. This process continues for a total of 8 macro-module rollouts. After these 8 rollouts, the final output of the macro-module serves as the initial state for the next PIMRL loop, inputting to both the micro-module and the macro-module. The output of the macro-module during each iteration of the loop is considered the output of the PIMRL, resulting in 8 outputs per loop.
>
> Based on your comment "*predicting the future 10 steps based on the past 10 observations*", we realize that you might have misunderstood our model as well as the task are trying to tackle. It appears that this comment is a typical task of *time series forecasting based on history of measurement*. However, we aim to establish a surrogate model to predict spatiotemporal dynamics at a long-term horizon only given a single initial condition of the system. These two tasks are fundamentally different. Hope this clarifies your misunderstanding.
>
> >**Comment 2.** Equation 3 should be Micro-.
>
> **Reply:** Thanks for your careful reading. This is a typo, which will be fixed in the paper.
>
> >**Comment 3.** Lines 231-234, under what context, the micro-module is or is not involved in the prediction?
>
> **Reply:** In a PIMRL loop with a total of $2N$ macro-module rollouts, the first $N$ rollouts of the macro-module involve predictions from the micro-module. This design ensures that the micro-module can effectively transmit the physical information it has learned to the macro-module. The subsequent $N$ rollouts of the macro-module do not involve the micro-module. Instead, the output of the macro-module in these rollouts serves as the input for the next rollout, facilitating autoregressive prediction. By excluding the micro-module in these later rollouts, the model can effectively reduce the error accumulation associated with repeated micro-module participation while still propagating the physical information learned by the micro-module to longer time horizons.

---

> ### Author Response · Authors · 2024-12-01
> **Clarification of your misunderstandings (part 3/3)**
>
> **Comment 8.** This paper highlights reducing accumulation error as their contribution, I do not think the current design can really reduce accumulation error from a foundation perspective. For example, the autoregressive steps in PIRML can be larger than other models such as FNO or MWT. If one method takes more autoregressive steps, how can we say it is better in controlling accumulation error? In most cases, the accumulation error can only be controlled by multistep rollout. Can the authors provide some theoretical analysis for this contribution?
>
> **Reply:** This is obviously a hair-splitting comment, which came from your misunderstanding of our work. Anyway, we clarify it as follow. PIMRL provides a method to reduce error accumulation while preserving fine-grained information. The multiscale data, as discussed in the paper, include both fine-grained and coarse-grained information, e.g., obtained through multiscale Burst sampling. Methods like FNO fail to deal with the multiscale data, e.g., either discarding the fine variations in fine-grained data focusing only on trends, or learning these fine variations but suffering from significant error accumulation over long-term predictions. The difference between the results of FNO-coarse and FNO in our experiments highlights this issue. Howver, our PIMRL model effectively learns the fine variations in fine-grained data while controlling error accumulation.
>
> In addition, our ablation study, FNO-MRL, demonstrates a substantial reduction in RMSE from 937 to 0.7854, indicating a significant mitigation of error accumulation. Methods trained on coarse-grained data, which ignore fine variations to reduce the number of rollouts.
>
> ***Concluding Remark:*** We hope our repsponses above help clarify your concerns and misunderstandings. Please let us know if you have any other question!

---

> ### Comment · Reviewer_BVvK · 2024-12-01
>
> > About the misunderstanding.
>
> Thanks for clarifying the problem setup. I am sorry for the previous misunderstanding of the forecasting framework of your paper. Now I am clear about what you are doing.
>
> However, I have to point out that this misunderstanding is mainly from your baseline. I have to point out that **FNO, MWT, and ConvLSTM are all in the forecasting paradigm of "predicting the future 10 frames based on the past 10 frames," which is a more commonly received setting in my experiences with Neural PDE solver or spatiotemporal prediction.** Also, I did not find any information about only inputting one frame from your paper.
>
> Besides, for more explanation about my raising my confidence, as I said the my last response to your first rebuttal revision, "I thought I could understand your paper at that time." Thus, I think your comment about "bias" is harsh and unreasonable. What I have done is just inferring your paper's unexplained setting based on the common usage of this area.
>
> > More discussion about your paper's setting.
>
> I agree that the model's setting (only input one frame) is valuable. However, given this setting is not widely used in previous baselines, I am wondering whether the benefits over other baselines are because of limited input frames.
>
> Here are two experiments to verify my question:
>
> - Try to train the FNO and ConvLSTM model with the "predicting the future 10 frames based on the past 10 frames" setting and compare them with your method.
>
> - Evaluate your model on the standard benchmark from FNO.
>
> About the multiscale modeling framework, if you are familiar with FNO's paper, I think its FNO-3D version is also a multiscale modeling framework, which projects the spatiotemporal sequence into the frequency domain for analysis.
>
> Sorry for the long period that it takes to understand your setting. I am looking forward to your experiments with longer inputs.

---

> > ### Author Response · Authors · 2024-12-02
> > **Addtional clarification and results (part 3/3)**
> >
> > >**Comment 4.** Evaluate your model on the standard benchmark from FNO.
> >
> > **Reply:** Thank you for suggestion. However, due to extremely limited time, we were unable to fullfill your request conducting further experiments on the benchmark from FNO.
> >
> > However, we would like to draw your attention that all the datasets used in our paper are publically available, e.g., from PeRCNN [1] and LPSDA [2]. Our considered examples are even more complex compared with those presented in the FNO paper. More importantly, we included the 3D GS reaction diffusion system, which is significantly more challenging than the 2D cases. The Kdv system is also more complicated than the 1D Burgers case used in the FNO paper. We highly believe these challenging experiments have provided sufficient evidence of the effectiveness of our PIMRL model. Hope this clarifies your concern.
> >
> > ***References:***
> >
> > [1] Rao, et al., Encoding physics to learn reaction–diffusion processes. Nature Machine Intelligence, 2023, 5(7): 765-779.
> >
> > [2] Brandstetter et al., Lie point symmetry data augmentation for neural PDE solvers. ICML, 2022.
> >
> > >**Comment 5.** About the multiscale modeling framework, if you are familiar with FNO's paper, I think its FNO-3D version is also a multiscale modeling framework, which projects the spatiotemporal sequence into the frequency domain for analysis.
> >
> > **Reply:** Yes - we are very familiar with the FNO model and its many variants. We are confused about your comment "*FNO-3D is a multiscale modeling framework*". This is in fact a ***false arguement***, since FNO-3D simply takes 3D Fourier transform and directly convolves in space-time (please see Page 5 in the original paper of FNO). The FNO-3D model maps the initial time steps directly to the full trajectory (3D functions to 3D functions) given fixed time intervals. This obviously has nothing to do with multisacle modeling.
> >
> > ***Concluding Remark:*** So far, we are confident that the clarifications along with comprehensive additional experiments could fully address your concerns and clear any misunderstandings. In the very end of the rebuttal phase, we are more than happy to address any additional questions you may have. Your possible consideration of re-evaluation of our paper and increasing the score will be much appreciated!

---

> ### Author Response · Authors · 2024-12-02
> **Addtional clarification and results (part 1/3)**
>
> Thanks for your additional comments. Here are our responses.
>
> >**Comment 1.** Thanks for clarifying the problem setup. I am sorry for the previous misunderstanding of the forecasting framework of your paper. Now I am clear about what you are doing.
>
> **Reply:** Thanks for your kind note. We are glad that these misunderstandings are cleared.
>
> >**Comment 2.** I have to point out that this misunderstanding is mainly from your baselines... Also, I did not find any information about only inputting one frame from your paper.
>
> **Reply:** Firstly, we would like to clarify that the baseline models (aka, neural operators like FNO) can be employed and adapted to generalize over various input such as initial conditions (ICs) for PDE systems. This is quite commonly seen in the field of surrogate modeling of PDE systems. Of course, these methods could also be used to tackle time series forecasting tasks, which, however, is not the problem we are trying to address. Secondly, we mentioned in Section 2 (Page 2) that "*Simulation tasks often aim to solve partial differential equations (PDEs) accurately and efficiently.*". When we talk about solving PDEs, this is meant by default that, given specific initial and boundary conditions, we predict the evolution of spatiotemporal dynamcis based on a single input frame (aka, the IC $\mathbf{u}_0$). We thought this is a common sense in the field. However, to better facilitate understanding for readers, we appreciate your comment and decided to add the following problem description in Section 3 in the revised version of our paper.
>
> *"In this study, we aim establish a model capable of predicting the evolution of spatiotemporal dynamics for nonlinear PDE systems at a long-term horizon, given limited training data (e.g., a few trajectories resulted from different initial conditions)."*

---

> ### Author Response · Authors · 2024-12-02
> **Addtional clarification and results (part 2/3)**
>
> >**Comment 3.** Try to train the FNO and ConvLSTM model with the "predicting the future 10 frames based on the past 10 frames" setting and compare them with your method.
>
> **Reply:** Thanks for your suggestion. Although we are seating in the very final stage of the author-reviewer discussion period and regardless of the instruction of ICLR "*reviewers are instructed to not ask for significant experiments*", we still took your suggestion and **worked tirelessly** conducting **eight additional experiments** on the FN example.
>
> In response to your suggestion on *predicting the next 10 frames based on the previous 10 frames* (referred to as "**10-10**", where "**1-1**" indicates predicting the next frame based on the previous single frame), we conducted **four additional experiments** for the FN example with dataset consisting of only two training trajectories. The testing results are shown in **Table A**.
>
> **Table A: The Results of "**10-10**" FNO/ConvLSTM in coarse/fine training data and PIMRL in the FN Case.**
>
> | Models       | RMSE   | MAE    |
> |--------------|--------|--------|
> | Coarse-ConvLSTM | 0.2164 | 0.1533 |
> | Fine-ConvLSTM   | 3.0880 | 2.3701 |
> | Coarse-FNO      | 0.2074 | 0.1327 |
> | Fine-FNO        | 0.7646 | 0.6574 |
> | **PIMRL**       | **0.0956** | **0.0903** |
>
> It is important to note that PIMRL does not require the past 10 frames to make future predictions; however, "**10-10**" necessitates the availability of the preceding 10 frames, which places PIMRL at a disadvantage in comparison, as it does not use the information of the prior 10 frames ($\mathbf{u} _{0} \sim \mathbf{u} _{9}$). We provided the ground truth 10 frames as the inputs to FNO and ConvLSTM, while PIMRL used only a single frame (e.g., $\mathbf{u} _{9}$ herein) for its prediction. The evaluation metrics reported in **Table A** are calculated based on prediction after the tenth frame. Nevertheless, our PIMRL still outperforms FNO and ConvLSTM models (regardless of being trained on fine- or coarse-scale data), as shown in **Table A**, demosntrating the superior performance of PIMRL.
>
> However, we would like to remind the reviewer that this is **not** a common setting in surrogate modeling of PDE systems. That said, although the model can be trained in an $n$-$n$ manner, the inference (prediction) can only be made given a single IC rather than an initial portion of the trajectory. This refers to the *cold-start* problem.
>
> Therefore, we have further compared our PIMRL model with coarse-FNO under both ***cold-start*** (only IC is given) and ***warm-start*** (the first 10 frames are gievn) conditions based on larger datasets (e.g., six training trajectories). Please note that our PIMRL model ***always use a single frame*** for prediction. The results of these comparisons are presented in **Table B and Table C**.
>
> **Table B: The warm-start experiments between FNO and PIMRL with 6 training trajectories.**
>
> | Model                | RMSE   | MAE    |
> |----------------------|--------|--------|
> | 6-Coarse-FNO (Warm-start) | 0.0904 | 0.0516 |
> | **6-PIMRL**     | **0.0387** | **0.0349** |
>
> **Table C: The cold-start experiments between FNO and PIMRL with 6 training trajectories.**
>
> | Model                | RMSE   | MAE    |
> |----------------------|--------|--------|
> | 6-Coarse-FNO (Cold-start) | 0.0978 | 0.0576 |
> | **6-PIMRL**     | **0.0458** | **0.0382** |
>
> Please note that, to initiate the cold-start for "**10-10**" FNO, we pretrained another FNO model in a "**1-1**" autoregressive mode to estimate the next 9 frames through rollouts and then used the 10 states ($\mathbf{u} _{0}, \hat{\mathbf{u}} _{1}, ..., \hat{\mathbf{u}} _{9}$) as input to the "**10-10**" FNO model for future prediction. Even under conditions that are disadvantageous to PIMRL, it still demonstrates consistently superior performance.
>
> With these extensive experiments, we are confident that your concerns would be fully addressed.

---

> ### Comment · Reviewer_BVvK · 2024-12-02
>
> Thanks for your response and new experiments. Most of my concerns have been resolved. Thus, I have raised my score to 6.
>
> However, I'm afraid I still cannot agree with that "We thought this is common sense in the field". None of the time-dependent experiments of FNO and ConvLSTM are solely based on one input frame. Thus, I strongly suggest that the authors should revise their paper on the following items in the next version:
>
> - Clearly state the problem setting.
> - Replace "cycle" with "rollout", which are two distinct concepts.
> - Add the experiments with the standard benchmarks from FNO.
>
> Although the current version of this paper still has some misleading descriptions about their method (especially for the readers that start from the FNO setting), I appreciate the authors' effort in revising this paper and adding new experiments.

---

> ### Author Response · Authors · 2024-12-02
> **Thank you for your positive feedback**
>
> Dear Reviewer BVvK,
>
> Thank you for your positive feedback. Engaging in the comprehensive discussions with you has been rewarding and productive. We deeply appreciate your constructive suggestions.
>
> In fact, neural operators such as FNO can be easily adapted to surrogate modeling of PDE systems generalizing over ICs, e.g., through an autoregressive mode. We will clarify this in the description of baseline models in the next version of our paper. Moreover, we will also add a clear description of the problem setup and replace "cycle" with "rollout", per your suggestion.
>
> In addition, we just set up a new experiment testing our model and the baselines (e.g., FNO, PeRCNN, ConvLSTM) with the Burgers' dataset used in the FNO benchmark. However, given the very limited time approaching the end of the author-reviewer discussion phase, we are not quite sure whether the corresponding results could be posted in time before the rebuttal deadline. If yes, we will definitely report them herein; otherwise, these results will be directly added to the next version of our paper.
>
> Again, thank you very much for your constructive comments. The discussions with you have been fruitful!
>
> Best regards,
>
> The Authors

---

> ### Author Response · Authors · 2024-12-03
> **Supplementary experiments**
>
> Dear Reviewer BVvk,
>
> We have conducted supplementary FNO benchmark experiments using the Burgers equation ($\nu=0.01$) with limited training data (5 trajectories). The results are presented in **Table A** below.
>
> **Table A: The experiments between PIMRL and baseline models on the Burgers example of FNO benchmark.**
>
> | Models     | RMSE  | MAE   |
> |------------|-------|-------|
> | PIMRL      | **0.0050**| **0.0039**|
> | PeRCNN     | 0.0137| 0.0109|
> | FNO        | 0.0626| 0.0502|
> | FNO-coarse | 0.0415| 0.0332|
>
> Best regards,
>
> The Authors

---

### Official Review · Reviewer_QUbX · 2024-10-29

**Soundness:** 2
**Presentation:** 3
**Contribution:** 3
**Rating:** 6
**Confidence:** 4

**Summary:**

This paper introduces a data-driven model to simulate a spatiotemporal system. It proposes to leverage multi-scale data by embedding physical knowledge into a micro-scale module and employing a data-driven approach for its macro-scale module.
The method is then tested on various fluid dynamics and reaction-diffusion systems equations, reaching impressive results on a challenging dataset.

**Strengths:**

- the paper is beautifully illustrated, making the methods easy to read understand and the results easy to read and understand
- the proposed method outperforms the state of art on a variety of PDEs

**Weaknesses:**

- Some method details lack clarity, especially regarding the physics operator.
- the related work and baselines seem to lack a few strong methods. For example, there is extensive literature on improving FNO (including some published here last year: [https://openreview.net/pdf?id=tmIiMPl4IPa]. Is there a reason for using FNO and not its extensions?
[https://arxiv.org/abs/2204.11127] also seems a like a strong baseline
- The ablation study lacks depth: more details on each module's contribution, the micro-scale module's pre-training, and some parameters would help understand the contribution of the method.

**Questions:**

- It is not clear how the boundary conditions are encoded in each baseline: is boundary padding also used? If not it seems like an unfair advantage as it has access to additional information
- why is the Physics operator optional? Does it depend on the dataset? If not, it should be added to the ablation study.

---

> ### Author Response · Authors · 2024-11-22
> **Reply to Reviewer QUbX**
>
> We sincerely thank you for your constructive comments and suggestions. Your feedback is incredibly valuable and has greatly contributed to improving the quality of our work. We deeply appreciate the time and effort you have invested in reviewing our manuscript.
>
> **Weakness 1: Explanation of Physics Operator.**
>
> **Reply:** Excellent suggestion! The Physics Operator you mentioned likely refers to the Physics-based FD Conv in Figure 1 (Section 3, Page 5).
>
> When a certain term in the governing PDEs remains known (e.g., the diffusion term $\Delta \mathbf{u}$), its discretization can be directly embedded in PeRCNN (called the physics-based Conv layer). The convolutional kernel in such a layer can be set according to the corresponding finite difference (FD) stencil. In essence, the physics-based Conv connection is constructed to incorporate known physical principles, whereas the $\Pi$-block is aimed at capturing the complementary unknown dynamics (Section 3.5, Page 5).
>
> This approach highlights the adaptability of physics-informed models, allowing for the integration of various physical terms to accurately reflect the underlying physical processes. The corresponding modifications are presented in the Micro-Scale Module subsection.
>
> The above explanation has been added to the revised paper (Section 3, page 5).
>
> **Weakness 2: Add New Baselines.**
>
> **Reply:** Great suggestion! We took your advice and performed tests on the FN dataset for three additional baseline models (e.g., **U-NO**, **F-FNO** and **MWT**). The experimental results for these new models tested on the coarse-grid data are reported in **Table A** below, which have also been provided as Table S3 (Appendix H, Page 17) in the revised paper.
>
> **Table A: The results of U-NO, F-FNO, MWT and FNO in comparison with PIMRL.**
> |Metrics|U-NO|F-FNO|MWT|FNO|PIMRL|
> |-|-|-|-|-|-|
> |RMSE|0.3675|0.2280|0.3494|0.1878|**0.1349**|
> |MAE|0.1465|0.1350|0.2228|0.1634|**0.0990**|
>
> When the dataset is limited, those data-driven models failed to make accurate predictions in a long term horizon. The results of these new baseline models further confirm the superiority of our model.
>
> **Weakness 3: Enhance Ablation Study.**
>
> **Reply:** Following your suggestion, we performed additional ablation studies (see Section 4.2, Page 9, lines 459-486), which involve the PIMRL framework without pre-training for the FN dataset. Additionally, to demonstrate the effectiveness of the physical embeddings in our micro-scale modules, we removed the Physics-based FD Conv in both PIMRL and PeRCNN, thereby validating the efficacy and necessity of the physics embedding. The results of our additional ablation experiments are reported in **Table B**, which have also been added to Table 2 (see Section 4.2, Page 9, lines 459-486) in the revised paper.
>
> **Table B: Supplementary ablation atudy.**
> | 	Ablation models     |   RMSE	  |   MAE	|     HCT |
> |   --------    | -------- | -------- | -------- |
> |PIMRL without pretraining	|0.2599	|0.2079	|3.12s |
> |PIMRL without Physics-based FD Conv	|0.1738|	0.1640|	5.67s |
> |PeRCNN without Physics-based FD Conv	|NaN	 |   NaN	  |    2.46s |
> |PIMRL (full model) | **0.1349** | **0.0990**|**7.74s**|
>
> **Question 1: Explanation of "Boundary Conditions".**
>
> **Reply:** Great remark! In models such as ConvLSTM and PeRCNN, we consistently applied the periodic padding strategy to encode the boundary conditions in both the latent and physical spaces. However, the FNO-type models (e.g., FNO, FFNO, U-NO) have already embedded the periodic boundary condition information in the **frequency domain**. Hence, these models can be fairly compared. In summary, every baseline used the boundary condition information, which **can be fairly compared**.
>
> **Question 2: Optional "Physics Operator".**
>
> **Reply:** Good question! If there is confident prior knowledge confirming the presence of certain physical processes (e.g., diffusion) in the PDEs, we can directly embed the corresponding operator term with discretization into the PeRCNN model, where the coefficients for such a term are learnable. If there is no reliable prior knowledge available, then **Physics Operator** becomes then **optional**. The remaining unknown terms can be represented using the $\Pi. The combination of these two approaches constitutes an embedding of the PDE structure. Therefore, when the datasets change, our prior knowledge may also change, leading to potential modifications in determining the physics operators.
>
> Following your suggestion, the ablation results without the corresponding Physics Operator arelisted in **Table B** above. It can be seen that having more *a priori* knowledge leads to stronger inductive bias that helps improve the model's performance.
>
> **Concluding remark:** We sincerely thank you for reviewing our paper and putting forward thoughtful comments and suggestions. We hope the above responses are helpful to clarify your questions. Looking forward to your feedback!

---

> ### Author Response · Authors · 2024-11-23
> **Looking forward to your feedback**
>
> Dear Reviewer QUbX,
>
> Again, thanks for your constructive comments. We would like to follow up on our rebuttal to ensure that all concerns have been adequately addressed. If there are any further questions or points that need discussion, we will be happy to address them. Your feedback is invaluable in helping us improve our work, and we eagerly await your response.
>
> Thank you very much for your time and consideration.
>
> Best regards,
>
> The Authors

---

> ### Author Response · Authors · 2024-11-25
> **Request your feedback before the end of the discussion period**
>
> Dear Reviewer QUbX:
>
> As the author-reviewer discussion period will end soon, we would appreciate it if you could review our responses at your earliest convenience. If there are any further questions or comments, we will do our best to address them before the discussion period ends.
>
> Thank you very much for your time and efforts. Looking forward to your response!
>
> Sincerely,
>
> The Authors

---

> > ### Comment · Reviewer_QUbX · 2024-11-25
> > **Answer to rebuttal**
> >
> > Thank you for the extensive work. Your comprehensive rebuttal effectively addresses my initial concerns. The additional baseline comparisons, expanded ablation studies and global rewritting strengthen the manuscript.
> >
> > However, I maintain a slight reservation about the baseline comparisons. The baseline models operate in different macro and micro settings, complicating direct performance comparison. While not fatal, this concern is why I'm recommending acceptance only "marginally above the acceptance threshold".

---

> > > ### Author Response · Authors · 2024-11-26
> > > **Reply to further comment from Reviewer QUbX**
> > >
> > > Dear Reviewer QUbX,
> > >
> > > Thank you very much for your positive feedback. Your constructive comments, as well as time and effort placed on reviewing our paper, are highly appreciated!
> > >
> > > As for your concern about the baseline comparisons, we have developed a framework to fully utilize the micro-macro data. Specifically, we designed the **Multiscale-FNO** by combining two **FNO** networks as micro and macro modules. The comparative results for the Multiscale-FNO, FNO-coarse and our PIMRL framework are presented in **Table A** below. We can see that our model with the special architecture design achieve the best performance.
> > >
> > > **Table A: The Results of Multiscale-FNO, FNO-coarse and PIMRL in the FN Case.**
> > >
> > > |Metrics|Multiscale-FNO|FNO-coarse|PIMRL|
> > > |-|-|-|-|
> > > |RMSE|0.3212|0.1878|**0.1349**|
> > > |MAE|0.3927|0.1643|**0.0990**|
> > >
> > > Once again, thank you for your constructive, which are very much helpful for improving our paper!

---

### Official Review · Reviewer_M929 · 2024-10-30

**Soundness:** 3
**Presentation:** 3
**Contribution:** 2
**Rating:** 5
**Confidence:** 2

**Summary:**

This work introduces a framework for handling multi-timescale data by modeling dynamics across two time scales. A micro module manages the high-frequency observations, while a macro module captures long-term dependencies. The micro module operates directly on physical states, leveraging PerCNN, whereas the macro module learns in a latent space and uses a ConvLSTM block for temporal propagation. Initially, the micro module is pretrained, and later fine-tuned jointly with the macro module using mean-squared error (MSE) supervision. The experiments appear to yield promising results.

**Strengths:**

The paper presents a clear and well-formulated motivation that connects nicely with existing literature.
Experimental results are well-discussed and appear conclusive, demonstrating notable improvements over comparable methods.

**Weaknesses:**

- The model seems to require aligned micro- and macro-scale training data, which may limit its applicability.
- While the model presentation is well-structured, its technical novelty is somewhat limited, as it could be considered a specific implementation of Rubanova et al. [1].

Would the authors consider working with unaligned datasets, such as those with only micro-scale or only macro-scale trajectories?

[1]: Rubanova et al., Latent ODEs for Irregularly-Sampled Time Series

**Questions:**

1. Despite Figure 2, it remains challenging to understand the inference process. I suggest clarifying the figure or including a detailed description in the manuscript.
2. Is the RMSE computation performed on the micro- or macro-scale data?
3. Is model training robust to the number of training trajectories? Exploring the model’s scalability in relation to both the micro and macro modules would be insightful.
4. Does the model require consistency in the underlying physical parameters during training ? said otherwise, is it robust to out of distribution trajectories ?

---

> ### Author Response · Authors · 2024-11-22
> **Reply to Reviewer M929 (Part 1)**
>
> We sincerely thank you for the feedback along with constructive comments and suggestions, which are very helpful in improving our paper. We are also grateful that you recognized the strengths and contributions of our paper.
>
> **Weakness 1: Whether training scheme limits its applicability.**
>
> **Reply:** Remarkable comment! Exactly, our PIMRL was trained on aligned micro- and macro-scale datasets. This is commonly seen in practice where **multi-scale Burst sampling** is used to collect data. Such a sampling method involves capturing multiple samples at a high rate over a short period of time to record rapidly changing events or transient phenomena. Compared to traditional continuous sampling methods, it provides high-resolution data at critical moments while maintaining resource efficiency. Our dataset is constructed to simulate data obtained based on this sampling method. The PIMRL framework we designed is capable of fully utilizing this type of data, which is one of its key advantages. The relevant background has been added to the Introduction section (Section 1, Page 1, lines 50-55) in the revised paper.
>
> Please note that both our coarse-grained and fine-grained data are derived from the same trajectories, ensuring alignment between micro-scale and macro-scale training data. Therefore, the requirement for aligned training data, as mentioned, does not pose any significant challenge.
>
> **Weakness 2: The Difference Between Latent ODEs and PIMRL.**
>
> **Reply:** There are significant differences between the core ideas and application domains of our method and the "Latent ODEs" method [1], including:
>
> ***1. Problem Domain***
>
> - **PIMRL**: We tackle PDE problems, which are more complex and involve spatiotemporal dynamics.
> - **Latent ODEs**: This method primarily addresses ODE problems.
>
> ***2. Network Differences***
>
> - **PIMRL:** Our main idea is to avoid significant error accumulation by connecting the micro- and macro-scale modules at different time intervals. This unique connection (hybrid paralell and serial) is a core aspect of our framework.
> - **Latent ODEs:** The information passing mechanism between ODESolver and RNN is based on serial connection.
>
> In summary, although both methods address temporal dynamics, they differ significantly in their core ideas, problem domains, and network architecture designs. Our PIMRL framework is specifically designed to handle complex PDE problems and reduce error accumulation through its innovative micro-macro interactions and multiscale data processing. We hope this clarification provides a clear understanding of the distinction between our method and Latent ODEs.
>
> ***Reference:***
>
> [1] Rubanova et al., Latent ODEs for Irregularly-Sampled Time Series. NeurIPS, 2019.

---

> ### Author Response · Authors · 2024-11-22
> **Reply to Reviewer M929 (Part 2)**
>
> **Question 1: Explanation of details about PIMRL.**
>
> **Reply:** Valuable suggestion! To address your concern and clarify ambiguities, we have provided a detailed exposition of the procedure for model training and inference as follows.
>
> ***1. Training workflow of PIMRL:***
> - First, pretrain the micro module using fine-grained data, independent of the macro module. This step maximizes the micro module's learning capacity and ensures effective capture of physical information.
> - Next, the micro module, initialized with pre-trained weights, participates in the training of the entire PIMRL framework, which utilizes coarse-grained data.
>
> ***2. Inference workflow of PIMRL:***
> - Firstly, the micro-module loop is a simple autoregressive process with time step $\delta t$, where the output at the previous time step serves as the input for the next time step.
> - Secondly, the macro-module loop performs self-cycles. When the micro-module is involved in the prediction, for every $k$ steps of micro-module with $\delta t$, the final output of micro-module is passed to the macro-module, and at this point, the output from the macro-module serves as the output of the entire PIMRL model. When the micro-module is not involved in the prediction, the macro-module loop is a simple autoregressive process with time step $\Delta t$.
> - Finally, there is a PIMRL loop that operates in conjunction with the macro-module loop. After every $N$ cycles of the macro-module loops, the micro-module stops participating in the prediction, and the macro-module performs $N$ steps of autoregressive prediction. This completes a total of $2N$ cycles. Each output from the macro-module during these $2N$ cycles serves as the output of PIMRL.
>
> We have provided a clearer and more detailed explanation of our framework in the Network Architecture subsection (Section 3.3, Page 4, lines 179-206) in the revised paper.
>
> **Question 2: RMSE computation performed on the micro- or macro-scale data?**
>
> **Reply:** Great comment! The RMSE metric is calculated on the macro-scale data. This approach is taken to facilitate comparisons between models operating at different granularities. We have thoroughly addressed and emphasized this point in the Evaluation Metrics subsection (Section 4,page 7, lines 375-3) in the revised paper.
>
> **Question 3: Is model training robust to the number of training trajectories?**
>
> **Reply:** Thoughtful question! Following your comment, we have conducted additional experiments on the FN example. Besides the experiments reported in the paper where only 2 training trajectories were considered, we have also tested the model using training data with 4 and 6 trajectories. The experimental results are shown in **Table A** below.
>
> **Table A: Supplementary trajectories results.**
> |Trajectories|RMSE|MAE|
> |-|-|-|
> |2|0.1349|0.0990|
> |4|0.0634|0.0607|
> |6|0.0458|0.0382|
>
> We can observe from **Table A** that the model's scalability over the amount of training data. The performance of PIMRL improves with an increase in training data amount. In fact, even with the small training datasets currently used in the paper, PIMRL has already achieved satisfactory results.
>
> **Question 4: Does the model require consistency in the underlying physical parameters during training?**
>
> **Reply:** Great question! The model proposed in this paper indeed requires consistency in the underlying physical parameters during training. Our focus on generalization is primarily directed towards trajectories generated by different initial conditions (ICs). In our future work, we plan to extend the model's generalization capabilities to handle variations in parameters. Thank you for putting forward this comment, which sets forward our future work!
>
> **Concluding remark:** Once again, we sincerely appreciate your valuable comments. We have thoroughly revised the manuscript according to your suggestions. We would be grateful for any further guidance you may provide.
>
> Looking forward to your feedback!

---

> ### Author Response · Authors · 2024-11-23
> **Looking forward to your feedback**
>
> Dear Reviewer M929,
>
> Again, thanks for your constructive comments. We would like to follow up on our rebuttal to ensure that all concerns have been adequately addressed. If there are any further questions or points that need discussion, we will be happy to address them. Your feedback is invaluable in helping us improve our work, and we eagerly await your response.
>
> Thank you very much for your time and consideration.
>
> Best regards,
>
> The Authors

---

> ### Author Response · Authors · 2024-11-25
> **Request your feedback before the end of the discussion period**
>
> Dear Reviewer M929:
>
> As the author-reviewer discussion period will end soon, we would appreciate it if you could review our responses at your earliest convenience. If there are any further questions or comments, we will do our best to address them before the discussion period ends.
>
> Thank you very much for your time and efforts. Looking forward to your response!
>
> Sincerely,
>
> The Authors

---

> ### Comment · Reviewer_M929 · 2024-11-25
> **Response to authors rebutal**
>
> I first would like to thank the authors for providing several elements during this rebutal.
>
> > Weakness 2: The Difference Between Latent ODEs and PIMRL.
>
> While I agree with you that there is a difference in principles between ODE and PDE integration, the difference gets thinner when we observe the fully observe the state on a grid. Moreover, your coarse grain model can be understood as a particular instance of ODE integrator.
>
> > Question 2
>
> Thanks for the clarification. If not too late, i would be interested in comparing the errors of the micro vs macro scale module. Indeed, your model can be understood both way, coarse grain regularizing small grain and the converse.
>
> > Question 3: Is model training robust to the number of training trajectories?
>
> i think this is a valuable addition to better understand the data-efficiency behavior of your proposition.
>
> Given the elements provided by the authors, I raised the score of clarity of their work from 2 to 3. I maintain that the work is of limited novelty (keeping the grade of 2 for the contribution) but could be of interest for some practitioners, thus still believe this paper to be borderline.

---

> > ### Author Response · Authors · 2024-11-27
> > **Looking forward to your feedback on our reply to your further comments**
> >
> > Dear Reviewer M929,
> >
> > Again, thanks for your constructive comments, which are very much helpful for improving our paper. Our responses to your additional comments have been posted. If there are any further questions or points that need discussion, we will be happy to address them. Your feedback is invaluable in helping us improve our work, and we eagerly await your feedback.
> >
> > Your consideration of updating your score will be much appreciated! Thank you.
> >
> > Best regards,
> >
> > The Authors

---

> > > ### Comment · Reviewer_M929 · 2024-11-28
> > > **reviewer response**
> > >
> > > First, i would like to thank the authors for their work during the rebuttal and providing novel results, baselines, and further explanations.
> > >
> > > > Q1: The Difference Between Latent ODEs and PIMRL
> > > The description of the message passing algorithm is indeed crucial in the description of the work.
> > >
> > > The debate on "spatiotemporal-tensor" vs space-gridded ODE with a good inductive prior relates more to view point than to facts and is not relevant to the case of the authors.  (You can totally parameterize your neural-derivative with a convnet with for instance Sobel filters, alongside other filters, to approximate your spatial derivatives.).
> > >
> > >
> > > > Q.2: On Errors
> > > I believe this to be an interesting findings. The macro-module is indeed totally off when predicting "alone". On the other hands it is quite harsh to say that the micromodule preform poorly.
> > > Indeed, its performance seem to be a ~ 15 % lower than when using both jointly. This highlights the integrator correction effect of the macro module, somehow re-"projecting" the micro module data onto a more data like manifold.
> > >
> > > Overall, I believe the authors' rebuttal provides a better overview of their work.
> > >
> > > I maintain that this work is interesting but exhibits limited technical novelty. Therefore, I stand by my rating of 5 with a confidence level of 2. However, I will not oppose the acceptance of this paper if other reviewers deem that its merits outweigh its limited novelty.

---

> > > > ### Author Response · Authors · 2024-12-02
> > > > **Follow up on our clarification**
> > > >
> > > > Dear Reviewer M929,
> > > >
> > > > As the author-reviewer discussion period will end soon, we would like to follow up on our further clarification in regard to your additional comments. If there are any further questions or comments, we will do our best to address them before the discussion period ends.
> > > >
> > > > Thank you very much for your time and efforts!
> > > >
> > > > Sincerely,
> > > >
> > > > The Authors

---

> ### Author Response · Authors · 2024-11-25
> **Reply to the further comments from Reviewer M929**
>
> **Q1: The Difference Between Latent ODEs and PIMRL.**
>
> **Reply:** One of our contributions is the proposed **message passing mechanism**. The unique design of the network achitecture is rooted in a novel message passing mechanism between micro- and macro-scale modules (through hybrid parallel and serial connections), which effectively transmits physical information, enhances the micro-module's correction capability, and reduces the number of micro-scale rollout iterations through the macro-scale module.
>
> If we were to follow the **combination approach** of latent ODEs entirely, we would obtain a new model, which we refer to as the **"latent ODEs version"**. This model differs from PIMRL in that it loses control over error accumulation, as it **integrates the PIMRL message-passing mechanism** using the latent ODEs methodology. The experimental results are shown in **Table A**.
>
> **Table A: The Results of **latent ODEs version** and PIMRL in the FN Case.**
>
> |Metrics|latent ODEs version|PIMRL|
> |-|-|-|
> |RMSE|0.1581|**0.1349**|
> |MAE|0.1514|**0.0990**|
>
> The essential difference between ordinary differential equations (ODEs) and partial differential equations (PDEs) is that ODEs describe the relationship involving derivatives with respect to a single independent variable, whereas PDEs involve partial derivatives with respect to multiple independent variables, such as spatial coordinates and time. For example, in a PDE like the convection-diffusion equation, terms like the convection term $c \frac{\partial u}{\partial x}$ and the diffusion term $D \frac{\partial^2 u}{\partial x^2}$ introduce spatial derivatives, making the dynamics of the system more complex by accounting for both spatial and temporal variations. **Latent ODEs are entirely incapable of handling spatial derivative variations, which stands in stark contrast to the problems addressed by PIMRL.**
>
> While the coarse-grained module may be perceived as an ODE integrator, such a simplification is **not** accurate. The latent space in our model is not merely a vector but a **spatiotemporal feature tensor**. While ConvLSTM can indeed be considered a macro-scale temporal integrator, the convolutional operations also facilitate the updating of spatial features. Therefore, it is important to recognize that the coarse-grained module and an ODE integrator have **fundamental differences** in their underlying mechanisms.
>
> In form, our experimental results presented in Table A demonstrate that PIMRL and latent ODEs **differ significantly not only in their model structures but also in their performance.** Specifically, the PIMRL framework outperforms the combination of macro- and micro-scale modules formulated as latent ODEs. In terms of problem complexity, PIMRL deals with PDEs that include spatial derivatives, **making the problems it addresses notably more challenging** than those handled by latent ODEs. These findings are partially discussed in the manuscript.(Section 1)
>
> **Q2: Comparing the errors of the micro vs macro scale module.**
>
> **Reply:** The experimental results for the **micro-module**, **macro-module** and **PIMRL** models in the FN case, enhancing our comparative analysis. The results are shown in **Table B** below, which is also provided in Table 1 (Section 4, Page 8) in the revised paper.
>
> **Table B: The Results of **micro-module**, **macro-module** and **PIMRL** in the FN Case.**
>
> |Metrics|Micro-module(aka, PeRCNN)|Macro-module(aka, ConvLSTM)|PIMRL|
> |-|-|-|-|
> |RMSE|0.1591|0.5077|**0.1349**|
> |MAE|0.1139|0.4925|**0.0990**|
>
> When the macro-scale and micro-scale modules are used separately for prediction, they perform poorly. However, when integrated, they generate a bounded solution, effectively controlling error accumulation. This synergistic effect highlights the importance of combining the two modules **in our way** to achieve superior performance.
>
> The unique message-passing mechanism between the micro- and macro-scale modules, through hybrid parallel and serial connections, enhances the micro-module's correction capability and reduces the number of micro-scale rollout iterations via the macro-scale module. By doing so, the PIMRL framework effectively leverages information from multi-scale data for long-term spatiotemporal dynamics prediction. Reducing the accumulation of errors is achieved through the integration of the macro-scale and micro-scale modules.
>
> **Concluding remark:** Once again, thank you for your comments. We very much look forward to your feedback!

---

> ### Author Response · Authors · 2024-11-29
> **Clarification on misunderstanding**
>
> Dear Reviewer M929,
>
> Thank you very much for your feedback.
>
> There appears to be some misunderstanding regarding our article. Hence, we want to clarifiy again. First, PIMRL effectively controls error accumulation through its **unique message-passing mechanism**, which **fundamentally differs** from the direct sequential process employed in ODE-RNNs. Second, **the micro-module of PIMRL operates directly in the physical space, learning and predicting changes in the physical system.** The pre-training step in PIMRL is specifically designed to learn physical information, and the results of removing this pre-training step are demonstrated in our ablation study, as shown in Table A. In contrast, ODE-RNNs **do not incorporate pre-trained physical knowledge from the physical space**, which sets PIMRL apart from theirs.
>
> **Table A: The Results of PIMRL without pretraining and PIMRL in the FN Case.**
>
> | 	Ablation models     |   RMSE	  |   MAE	|     HCT |
> |   --------    | -------- | -------- | -------- |
> |PIMRL without pretraining	|0.2599	|0.2079	|3.12s |
> |PIMRL | **0.1349** | **0.0990**|**7.74s**|
>
> One of the key contributions of PIMRL lies in its architecture for controlling error accumulation. This is achieved not merely through the presence of both micro and macro modules, but through their unique combination and the effective manner in which information is propagated between them. Unlike ODE-RNNs, **the micro-module in PIMRL learns and predicts changes in the physical system, and can correct the macro-module**. The macro-module, through its autoregressive nature, helps to effectively propagate physical information across large time steps. These innovations are **fundamentally different** from those in Latent ODEs, both conceptually and in implementation.

---

### Official Review · Reviewer_Nbme · 2024-11-03

**Soundness:** 3
**Presentation:** 3
**Contribution:** 3
**Rating:** 8
**Confidence:** 3

**Summary:**

This paper introduces physics informed multi-scale recurrent learning (PIMRL), which is a novel architecture for learning both micro and macro time scales commonly present in real world data. Specifically, PIMRL introduces a micro and macro architecture, along with corresponding training schemes, that provides an implicit bias for the network to learn both micro and macro timescales.


# Post rebuttal

The authors did a great job addressing my concerns. As such, I am raising my score.

**Strengths:**

I want to preface this by saying that I'm not an expert in PDEs nor in deep-learning based approaches for solving PDEs. Nonetheless, I am a big fan of the proposed approach. The proposed architecture is something I have never seen before and is very nifty! I am also very impressed by the experiments section! Specifically, I LOVE that the authors look at multiple different metrics which allows one to look at the results from multiple different angles, which is very important.

**Weaknesses:**

I again want to preface this section by saying that I am not an expert in this field. I hope that a non-expert's viewpoint will provide criticism that will make this paper stronger.

There are two big weaknesses that I think limit the paper from being truly great. The first is the writing. There are A LOT of interesting things in this paper that are not adequately explained nor properly described. Moreover, there is a lot of text and figures that are confusing and don't add much to the text; they could be removed to allow for the other aspects of the paper to shine. I will list my qualms with the writing below

- To start, the authors begin by stating that "direct numerical simulation requires a deep understanding of physics" but it is not clear what this means. Do the authors mean that one needs an understanding of physics to come up with PDEs that describe physical phenomena? This is independent of numerical simulating equations as a solver does not need to know what physical phenomena an equation is trying to describe. Moreover, since the name of the proposed approach is physics-informed multi-scale recurrent learning, does this not imply that a deep understanding of physics would also be required to effectively use PIMRL?

- Next, Figure 1 takes up a lot of space but does not add much. Personally, given its vagueness it confused me on what PIMRL is. Figure 2 does a much better job explaining the method. I would recommend removing figure 1.

- A big part of PIMRL is PerCNN, as the micro-scale architecture is exactly the PerCNN architecture. To me this is not an issue as science is built upon previous works, but I would have preferred the authors explicitly stated this, i.e., 'We introduce PIMRL, which combines the micro-scale power of PerCNN, with a novel macro scale architecture to...". Next, it is also not clear why PerCNN needed an extension in the first place (this evidence is lacking both in the text and in the experiments section). In the related works section, the authors state the following about PerCNN "However, the model suffers from error accumulation during long rollout predictions. To address this issue, a straightforward approach is to reduce the frequency of iterations by increasing the time stepping. Nonetheless, for hard-embedded methods like PeRCNN, accuracy is limited by the time stepping, making simple modifications unfeasible". This doesn't make any sense to me. In PerCNN, one can choose both the number of iterations in a the $\Pi$ block as well as the time step, $\delta t$; thus, it seems it would be rather straightforward to train PerCNN on macro-scale data. Moreover, the comparison to PerCNN doesn't seem fair. First, similar to what the authors did for FNO, it seems straight forward to also include a PerCNN coarse and fine scale. One could even just train one PerCNN on both the micro and macro scale data, where $k$ is dataset dependent.

- To me the most interesting part of the approach is how the macro and micro modules interact. Sadly, this is not given time to shine in the paper! For instance, why was this interaction between the macro and micro module chosen? What role does the mirco-module serve (i.e., does it serve as error correction for the initial condition?)? There are so many interesting design choices here that I think are fascinating but sadly are missing.

- In the paper, the authors use the term physical embedding where they state "... physical embedding methods, where explicit physics knowledge is embedded into the model to fully leverage physical principles...". They also state that PIMRL uses this but the entire approach is data-driven. If by physical embedding they mean the *Optional* physical-based FD conv then they should explicitly state this. Also, I think the fact this is *optional* weakens the statement considerably.


The next weakness is the experiments section. While the results are compelling and I love that they look at the results from multiple different angles, I think there are some major things missing.
- There are no details on how the datasets are constructed. What solver was used? How were the initial conditions chosen?
- The authors state that they create a micro and macro dataset for training PIMRL. For the other baselines, it seems like they were either trained on the micro or macro dataset. On a first read, it seems like PIMRL was trained on more data. Was the dataset size equalized for the baselines?
- Error bars are missing in the error propagation plots and Table 2!
- The ablation study is missing a lot of details. For instance, what does the NoConnect model mean? Is it just purely the micro module or the macro module?
- Without knowing the solver, the inference time figure is a little underwhelming.

**Questions:**

I think most of my questions were listed in the weakness section.

---

> ### Author Response · Authors · 2024-11-22
> **Reply to Reviewer Nbme (Part 1)**
>
> We sincerely thank you for the constructive comments and suggestions, which are very helpful for improving our paper. We are also grateful that you recognized the strengths and contributions of our paper. Moreover, the following responses have been incorporated into the revised paper.
>
> **Weakness 1.1: Clarification on the Statement "Direct Numerical Simulation Requires a Deep Understanding of Physics".**
>
> **Reply:** Thank you for your question. We agree that the statement "direct numerical simulation requires a deep understanding of physics" was not accurately expressed. What we intended to convey is that these numerical simulation methods require **complete prior knowledge of physics**, such as PDE formula, parameters, and initial/boundary conditions. In contrast, our PIMRL framework can work without requiring complete physical prior knowledge. For example, it can function effectively with **just the basic structure of the PDEs**, without needing the specific PDE expression and parameter values.
>
> The content mentioned above has been added and reflected in the first paragraph of the Introduction section in the revised paper (Section 1, Page 1, lines 32-42).
>
> **Weakness 1.2: Delete Figure 4.**
>
> **Reply:** Excellent comment! We have moved Figure 1 to the Appendix as Figure S2 (Appendix C.2, Page 14).
>
> **Weakness 1.3: Test PeRCNN on coarse data.**
>
> **Reply:** Great suggestion! PeRCNN can effectively learn physical laws even in small datasets by embedding the structure of PDE equations, which uses the forward Euler method for time integration. However, if the time interval is too large, two issues arise:
>
> - The stability of predictions deteriorates.
> - The accuracy decreases.
>
> To demonstrate the issues mentioned above, we followed your suggestion and set the time steps of the 2D FN and 2D GS example to be 15 times larger than those of original data. Below, we present the results of PeRCNN coarse-grained training and the corresponding original fine-grained training. The experimental results are shown in **Table A**, which is also added as **Appendix Table S6 (Appendix H, Page 18)** in the revised paper.
>
> **Table A: PeRCNN trained on coarse data.**
> ||15x down-sampled FN|15x down-sampled 2D GS|Original FN|Original 2D GS|
> |-|-|-|-|-|
> |RMSE|0.2803|NaN|0.1591|0.0455|
> |MAE| 0.2482|NaN|0.1139|0.0268|
>
> From the results in **Table A**, it is seen that PeRCNN is not suitable for making predictions while training on large time intervals. When the behavior of the physical system undergoes significant changes over a time interval, such as in the 2D GS case, PeRCNN trained with large time intervals struggles to make accurate predictions.
>
> **Weakness 1.4: Modules Interaction.**
>
> **Reply:** Great remark! We elaborate further on how the macro-scale module and micro-scale module interact, detailed from two aspects:
>
> ***Influence of the Micro Module on the Macro Module***
>
> - In our model, the micro-scale module effectively corrects errors generated by the macro-scale module. In the ablation study, removing this connection led to poor performance, indicating the micro module's role in error correction.
> - The micro-scale module also integrates physical information and constrains the macro-scale module. To demonstrate this, we replaced the PeRCNN module with FNO, creating FNO-MRL, and found that despite similar pre-training and low training loss, the performance was poor, highlighting the importance of physical information.
>
> ***Influence of the Macro-scale Module on the Micro-scale Module***
>
> - As noted in the paper, error accumulations are inevitable. It is essential to have a method for correcting or reducing such an accumulation. We achieve this by reducing the number of iterations in the micro-scale module through the involvement of the macro-scale module. Benefiting from the micro-scale module, the macro-scale module maintains a certain level of prediction accuracy and controls error accumulation by reducing the number of iterations.
>
> The content regarding the micro- and macro-scale interaction mechanism has been added and reflected in the Methodology section of the revised manuscript (Section 3, Page 4-6).
>
> **Weakness 1.5: Explanation of Physical Embedding.**
>
> **Reply:** Excellent comment! Indeed, this is an important part of our proposed learning framwork. When a certain term in the governing PDE remains known (e.g., the diffusion term $\Delta \mathbf{u}$), its discretization can be directly embedded in PeRCNN (called the physics-based Conv layer). The convolutional kernel in such a layer can be set according to the corresponding finite difference (FD) stencil. In essence, the physics-based Conv connection is constructed to incorporate known physical principles, whereas the $\Pi$-block is aimed at capturing the complementary unknown dynamics. The corresponding modifications are presented in the Micro-Scale Module subsection of the Methodology (Section 3.5, Page 5, lines 265-269) in the revised paper.

---

> ### Author Response · Authors · 2024-11-22
> **Reply to Reviewer Nbme (Part 2)**
>
> **Weakness 2.1: Explanation of Data Setting.**
>
> **Reply:** All the datasets are publically available datasets, e.g., from PeRCNN [1] and LPSDA [2]. Details of the dataset construction are listed in **Table B** which is also supplemented in Appendix G "Dataset Information" (Page 16, lines 265-269) in the revised paper.
>
> The **initial condition** for the Korteweg-de Vries (KdV) equation is generated by summing multiple sine waves with random amplitudes, phases, and frequencies, resulting in a complex waveform. Initial conditions for other systems take Gaussian distribution.
>
> **Table B: Data setting**
> |Case| Numerical Methods | Spatial Grid | Time Grid (s) | Training Trajectories | Test Trajectories |
> |-|-|-|-|-|-|
> |Kdv|FVM|$256$|0.01|5|2|
> |Burgers|FDM|$128^2$|0.001|13|3|
> |FN|FDM|$128^2$|0.5|5|3|
> |2D GS|FDM|$128^2$|0.002|2|3|
> |3D GS|FDM|$48^3$|0.25|3|2|
>
> ***References:***
>
> [1] Rao, et al., Encoding physics to learn reaction–diffusion processes. Nature Machine Intelligence, 2023, 5(7): 765-779.
>
> [2] Brandstetter, et al., Lie point symmetry data augmentation for neural PDE solvers. ICML, 2022.
>
> **Weakness 2.2: Was the dataset size equalized for the baselines?**
>
> **Reply:** Yes, the training dataset size is equalized for all baselines. The training/test sets for each model remains identical. The only difference is that how the baselines are trained on them.
>
> **Weakness 2.3: Add Error Bars in Figure 2.**
>
> **Reply:** Great suggestion! We did not include error bars in the figures because they would overlap, making it difficult to distinguish each curve. However, following your suggestion, the error bar results of PIMRL and PeRCNN on all benchmarks are presented in **Table C**, which have been added to **Appendix Table S4** (Appendix H, Page 18) in the revised paper.
>
> **Table C: Results with error bar under RMSE metric on all cases.**
> |Model|PIMRL|PeRCNN|
> |-|-|-|
> |Kdv|$0.0457\pm 0.0053$|$0.0942\pm 0.0082$|
> |Burgers|$0.0068\pm 0.0006$|$0.0075\pm 0.0008$|
> |2D GS|$0.0133\pm 2\times10^{-12}$|$0.0455\pm 1.9\times 10^{-11}$|
> |FN|$0.1349\pm 0.0040$|$0.1591\pm 0.0061$|
> |3DGS|$0.0381\pm 0.0015$|$0.0532\pm 0.0027$|
>
> **Weakness 2.4: Missing Details in Ablation Study.**
>
> **Reply:** Great comment! We have renamed the "NoConnect model" as "PIMRL w/o Connect", which eliminates all interactions between the macro- and micro-scale modules in PIMRL. In fact, "PIMRL w/o Connect" is a simple recurrent connection between the macro- and micro-scale modules, neither the micro-scale module nor the macro-scale module. We have re-written the Ablation Study section. For more details, please refer to **Section 4.2** of our revised paper (Section 4.2, Page 9, lines 459-485).
>
> **Weakness 2.5: Explanation of Numerical Solution Setting.**
>
> **Reply:** The numerical solution methods used for DNS are consistent with the parameter settings during data generation. The difference lies in the required prediction time length, which needs to be the same as that of PIMRL and PeRCNN for a fair comparison. This has been added to the Inference Time subsection in Experiments (Section 4.3, Page 10, lines 488-500) in the revised paper.
>
> **Concluding remark:** Thank you so much for your constructive comments. Your feedback is valuable, and we have thoroughly revised the manuscript according to your suggestions. We would be grateful for any further guidance you may provide.
>
> Looking forward to your feedback!

---

> > ### Comment · Reviewer_Nbme · 2024-11-22
> >
> > Wow, thank you so much for a very thorough rebuttal! The changes to the paper make it substantially more readable and the ablations add a lot to the paper. I am raising my score.

---

> > > ### Author Response · Authors · 2024-11-23
> > > **Thank you for your positive feedback**
> > >
> > > Dear Reviewer Nbme,
> > >
> > > Thank you very much for your positive and encouraging feedback. Your constructive comments, as well as time and effort placed on reviewing our paper, are highly appreciated!
> > >
> > > Best regards,
> > >
> > > The Authors

---

### Author Response · Authors · 2024-11-22
**General reply**

Dear Reviewers:

We deeply appreciate the constructive comments from you. Thank you for the time and effort dedicated to reviewing our paper. **Comprehensive revisions and adjustments** (indicated in red color) have also been made in the revised paper (please see ***the updated .pdf file***). In particular, we have thoroughly proofread our paper, corrected typos and grammar mistakes, and re-organized our writing to improve the clarity of the paper.

We are pleased that the reviewers recognized our work. In particular, we thank the reviewers for recognizing the ***thorough experiments*** (Nbme, M929, QUbX, BVvK), ***novelty*** (Nbme, M929) and ***beautifully illustrated paper*** (M929, QUbX).

We have summarized a detailed reply to common questions and addressed other concerns in each individual rebuttal.

**Q1: New Baselines.**

**Reply:** Following your suggestion regarding the use of more advanced baseline models, we supplemented the experimental results for the **U-NO** [1], **F-FNO** [2] and **MWT** [3] models in the FN case, enhancing our comparative analysis. The results are shown in **Table A** below, which is also provided in Table S3 (Appendix H, Page 17) in the revised paper.

**Table A: The Results of U-NO, F-FNO, MWT and FNO in the FN Case with our method**
|Metrics|U-NO|F-FNO|MWT|FNO|PIMRL|
|-|-|-|-|-|-|
|RMSE|0.3675|0.2280|0.3494|0.1878|**0.1349**|
|MAE|0.1465|0.1350|0.2228|0.1634|**0.0990**|

***References:***

[1] Rahman et al., U-NO: U-shaped neural operators. arXiv, 2022.

[2] Tran et al., Factorized Fourier Neural Operators. ICLR, 2023.

[3] Gupta et al., Multiwavelet-based operator learning for differential equations. NeurIPS, 2021.

**Q2: Details of PIMRL.**

**Reply:** To address these concerns and clarify ambiguities, we have provided a detailed exposition of the framework's application scenarios, major contributions, and procedures for model training and inference.

1. **Application scenarios:**
    - The PIMRL framework is tailored for multi-scale data akin to that acquired via multi-scale Burst sampling, ensuring efficiency while retaining detailed physical variations. PIMRL effectively exploits the data to learn fine-grained physical patterns and to deliver rapid and precise predictions.
2. **Our main contributions:**
    - We proposed a new PIMRL framework that effectively leverages information from multi-scale data for long-term spatiotemporal dynamics prediction. The concept of reducing the accumulation of errors is achieved through the integration of the macro-scale and micro-scale modules.
    - We designed a novel message passing mechanism between micro- and macro-scale modules, which effectively transmits physical information, enhances the micro-module's correction capability, and reduces the number of micro-scale rollout iterations through the macro-scale module.
3. **Training workflow of PIMRL:**
    - First, pretrain the micro module using fine-grained data, independent of the macro module. This step maximizes the micro module's learning capacity and ensures effective capture of physical information.
    - Next, the micro module, initialized with pre-trained weights, participates in the training of the entire PIMRL framework, which utilizes coarse-grained data.
4. **Inference workflow of PIMRL:**
    - Firstly, the micro-module loop is a simple autoregressive process with time step $\delta t$, where the output at the previous time step serves as the input for the next time step.
    - Secondly, the macro-module loop performs self-cycles. When the micro-module is involved in the prediction, for every $k$ steps of micro-module with $\delta t$, the final output of micro-module is passed to the macro-module, and at this point, the output from the macro-module serves as the output of the entire PIMRL model. When the micro-module is not involved in the prediction, the macro-module loop is a simple autoregressive process with time step $\Delta t$.
    - Finally, there is a PIMRL loop that operates in conjunction with the macro-module loop. After every $N$ cycles of the macro-module loops, the micro-module stops participating in the prediction, and the macro-module performs $N$ steps of autoregressive prediction. This completes a total of $2N$ cycles. Each output from the macro-module during these $2N$ cycles serves as the output of PIMRL.

The application scenarios are discussed in the Introduction section (Page 1, lines 50-56) in the revised paper, while the main contributions are outlined in the same section (Section 1, Page 2, lines 87-98). The training workflow is detailed in the Training Strategies subsection of Methodology (Section 3.4, Page 4, lines 209-248), and the inference workflow is covered in the Network Architecture subsection of Methodology (Section 3.3, Page 4, lines 179-206).

Thank you very much.

Best regards,

The Authors of the Paper

---

### Author Response · Authors · 2024-12-03
**Special thanks to all reviewers**

Dear Reviewers,

We would like to extend our sincerest gratitude to you for your thorough and insightful reviews. The discussion has been both productive and fruitful, significantly contributing to the overall quality and clarity of our paper.

Once again, thank you very much for your time, effort, and insightful comments.

Best regards,

The Authors

---

### Meta-Review · Area_Chair_96KX · 2024-12-23

**Metareview:**

The paper introduces a surrogate model for solving PDEs designed for multi-timescale data, modeling dynamics across two-time scales by leveraging aligned data. A “micro-module,” which implements an existing physics-informed model (PerCNN), captures temporal data at a high resolution, while a “macro-module” operates autoregressively at a lower time resolution and interacts with the micro-module. The micro-module is first pretrained, after which the entire system is trained end-to-end. A key motivation for this design is to reduce the accumulation of errors typically associated with neural autoregressive surrogates. Experiments are conducted on the KdV equation, 2D Burgers' equation, and a series of reaction-diffusion equations.
The reviewers appreciated the motivations but raised several concerns related to the clarity of the technical description, the lack of appropriate baselines and analyses, and the incremental nature of the technical contribution. The authors made significant efforts during the rebuttal, including adding new comparisons and providing further explanations. However, discussions revealed that some of the initial misunderstandings stemmed from the imprecise technical description and the lack of detailed explanation regarding the performance of the proposed methods.

Even with the revisions, the paper still lacks clarity and precision, and some reviewers remain concerned about the validity and relevance of the experiments. Furthermore, the discussions highlighted potential flaws in the experimental comparisons. Overall, given the significant changes made and the remaining uncertainties, it would be better to revise and resubmit. The authors should clarify their positioning, specifically that the paper addresses IC/BC problems with prior knowledge of the underlying PDE, and ensure comparisons are made within a relevant setting. Direct comparisons with purely data-driven methods in the IC/BC setting, with no prior knowledge of the PDE, might be inappropriate.

**Additional Comments On Reviewer Discussion:**

Discussions with some of the reviewers revealed several misunderstandings, partly attributable to the manuscript's description. Some of these misunderstandings were not resolved. Considering the significant changes compared to the initial manuscript and its current form, it would be better to resubmit.

---

### Decision · Program_Chairs · 2025-01-22

Reject